# p62/SQSTM1-droplet serves as a platform for autophagosome formation and anti-oxidative stress response

Shun Kageyama[1,14], Sigurdur Runar Gudmundsson [2,14], Yu-Shin Sou[3], Yoshinobu Ichimura[1], Naoki Tamura[4], Saiko Kazuno [5], Takashi Ueno[5], Yoshiki Miura[5], Daisuke Noshiro [6], Manabu Abe[7], Tsunehiro Mizushima[8], Nobuaki Miura [9], Shujiro Okuda [9], Hozumi Motohashi [10], Jin-A Lee [11], Kenji Sakimura[7], Tomoyuki Ohe [12], Nobuo N. Noda [6], Satoshi Waguri [4], Eeva-Liisa Eskelinen [2,13✉] & Masaaki Komatsu [1✉]

Autophagy contributes to the selective degradation of liquid droplets, including the P-Granule, Ape1-complex and p62/SQSTM1-body, although the molecular mechanisms and physiological relevance of selective degradation remain unclear. In this report, we describe the properties of endogenous p62-bodies, the effect of autophagosome biogenesis on these bodies, and the in vivo significance of their turnover. p62-bodies are low-liquidity gels containing ubiquitin and core autophagy-related proteins. Multiple autophagosomes form on the p62-gels, and the interaction of autophagosome-localizing Atg8-proteins with p62 directs autophagosome formation toward the p62-gel. Keap1 also reversibly translocates to the p62-gels in a p62-binding dependent fashion to activate the transcription factor Nrf2. Mice deficient for Atg8-interaction-dependent selective autophagy show that impaired turnover of p62-gels leads to Nrf2 hyperactivation in vivo. These results indicate that p62-gels are not simple substrates for autophagy but serve as platforms for both autophagosome formation and anti-oxidative stress.

[1] Department of Physiology, Juntendo University Graduate School of Medicine, Bunkyo-ku, Tokyo 113-8421, Japan. [2] Molecular and Integrative Biosciences Research Programme, University of Helsinki, Helsinki 00014, Finland. [3] Department of Cell Biology and Neuroscience, Juntendo University Graduate School of Medicine, Bunkyo-ku, Tokyo 113-8421, Japan. [4] Department of Anatomy and Histology, Fukushima Medical University School of Medicine, Hikarigaoka, Fukushima 960-1295, Japan. [5] Laboratory of Proteomics and Biomolecular Science, Research Support Center, Juntendo University Graduate School of Medicine, Bunkyo-ku, Tokyo 113-8421, Japan. [6] Institute of Microbial Chemistry (BIKAKEN), Shinagawa-ku, Tokyo 141-0021, Japan. [7] Department of Animal Model Development, Brain Research Institute, Niigata University, Niigata 951-8510, Japan. [8] Picobiology Institute, Graduate School of Life Science, University of Hyogo, 3-2-1, Kouto, Kamigori-cho, Ako-gun, Hyogo 678-1297, Japan. [9] Bioinformatics Laboratory, Niigata University Graduate School of Medical and Dental Sciences, Chuo-ku, Niigata 951-8510, Japan. [10] Department of Gene Expression Regulation, Institute of Development, Aging and Cancer, Tohoku University, Sendai 980-8575, Japan. [11] Department of Biological Sciences and Biotechnology, College of Life Sciences and Nanotechnology, Hannam University, Daejeon 34430, Korea. [12] Department of Pharmaceutical Sciences, Faculty of Pharmacy, Keio University, Minato-ku 105-8512 Tokyo, Japan. [13] Institute of Biomedicine, University of Turku, Turku FI-20014, Finland. [14] These authors contributed equally: Shun Kageyama, Sigurdur Runar Gudmundsson. ✉email: eeva-liisa.eskelinen@utu.fi; mkomatsu@juntendo.ac.jp

Macroautophagy (thereafter referred to as autophagy) is defined as intracellular protein degradation pathway in lysosome[1,2]. The initial step of autophagy is the development of a single membrane sac, called isolation membrane/phagophore, close to endoplasmic reticulum (ER). The isolation membrane/phagophore sequesters a part of cytoplasm, forming a double membrane structure, autophagosome[3]. The autophagosome fuses with a lysosome to degrade the sequestered materials into building blocks such as amino acids. There are two modes of autophagy: bulk and selective autophagy[4]. While the former is responsible for metabolic adaptation, the latter one contributes to cellular homeostasis[5]. Actually, mice defective in autophagy are sensitive to metabolic stress and also suffer from cell degeneration due to the accumulation of misfolded proteins and damaged organelles[6]. Both modes are driven by core Autophagy-related (ATG) proteins, but a definite difference among them is that either the autophagosome randomly sequesters cytoplasmic components or it forms along specific cargos[4]. Specific autophagic cargos such as liquid droplets, damaged or excess organelles and aggregated proteins appear in the cytoplasm and are subsequently tagged with ubiquitin, leading to assembly of adaptor proteins such as p62, NBR1, NDP52, OPTN, and TAX1BP1 that bind to the ubiquitin chain[7]. Alternatively, transmembrane type of adaptor proteins, such as NIX, FAM134, and TEX264, directly localize on cargos[7]. In the case of the transmembrane adaptors, molecular markers like ubiquitin are not needed. Core autophagy-related proteins such as FAK family kinase-interacting protein of 200 kDa (FIP200) also recognize the labelled targets through the interaction with adaptor proteins[8–10], beginning the process of autophagosome formation around the targets. Ubiquitin-binding and transmembrane type adaptor proteins, both have LC3-interacting region(s)/GABARAP-interaction motif(s), called LIR/GIM to interact with autophagosome-localizing ATG8 family proteins (including LC3A, LC3B, LC3C, gamma-aminobutyric acid receptor-associated protein (GABARAP), gamma-aminobutyric acid receptor-associated protein-like (GABARAPL)1, and GABARAPL2) around the cargos[11]. To date, gene targeting for Atg genes revealed various physiological functions of autophagy[12], but the membrane dynamics and physiological roles focusing on selective autophagy specifically are not well known. Herein, we investigated the properties of endogenous p62/SQSTM1 (thereafter referred to as p62)-structures, which are a representative of a selective substrate for autophagy. We clarified the membrane dynamics of isolation membranes/phagophores targeting the p62-structures as well as the in vivo significance of autophagic turnover of the p62-structures.

## Results

**p62-positive structures in cells are gels.** Recent studies revealed that liquid droplets, such as Ape1 complex, stress granule, P-body, and p62-body, which are formed by liquid–liquid phase separation, are selectively degraded by autophagy[13–19]. In the case of selective autophagy for Ape1 complex, P-body, and stress granule[13,16,20], liquidity or interacting protein(s) have been shown to be crucial factors for the structures to be surrounded by an autophagosome. Despite p62 is a typical selective substrate for autophagy and forms round structures with a range from 1 to 5 μm within cytoplasm[21], the dynamics of endogenous p62-structures still remains unknown. To investigate the properties and dynamics of endogenous p62-structures and their isolation by autophagosomes in cells, we utilized a hepatocellular carcinoma cell line, Huh-1, in which endogenous p62 expresses at high level[22]. Huh-1 cells possessed a large number of the p62-positive structures in the cytoplasm (Fig. 1a). Such structures were round

with a diameter from 0.5 to 3 μm and completely colocalized with the Neighbor of BRCA1 gene 1 protein (NBR1), a binding partner of p62[23] (Fig. 1a). p62 is phosphorylated at Ser403 (Ser405 in mice)[24] and Ser349 (Ser351 in mice)[25] to enhance the binding to ubiquitinated proteins and to recruit kelch-like ECH-associated protein 1 (Keap1) on the p62-structures, respectively. The p62-positive structures in Huh-1 cells contained both phosphorylated forms and were also positive for ubiquitin and Keap1 (Fig. 1a). Time lapse imaging of Huh-1 cells expressing GFP-p62 indicated that the p62-bodies have a spherical shape, move through the cytoplasm and occasionally fuse with each other (Supplementary Movie S1), matching with the criteria of liquid droplets and/or gels[26]. We next purified the p62-structures from GFP-p62 expressing Huh-1 cells and then treated them with 1,6-hexanediol, an aliphatic alcohol that has been used to differentiate between liquid droplets and aggregates[27]. The structures dispersed within 5 min after the treatment (Fig. 1b and Supplementary Movie S2). Fluorescence recovery after photobleaching analysis (FRAP) showed that average fluorescence recovery time after photobleaching of GFP-p62-labelled p62-structures is $9.2 \pm 1.2$ min (Fig. 1c and Supplementary Movie S3). Considering that the recovery times are much slower compared with that of typical liquid droplets (scales of seconds to minutes)[17,18,26], we concluded that the p62-structures in Huh-1 cells are gels.

**p62-gels are degraded by autophagy.** Core ATG proteins essential for autophagosome formation, such as FIP200, ULK1, WIPI2, and ATG16L1, were recruited to the p62-positive gels (Fig. 1d), implying autophagosome formation is constantly primed around the gels. Actually, treatment with BafilomycinA$_1$ (BafA$_1$), which is an inhibitor of lysosomal acidification, significantly increased the levels of p62, S349- and S403-phosphorylated p62 and NBR1 (Fig. 1e). These fluctuations were completely consistent with those of LC3-II and GABARAP-II (Fig. 1e). Likewise, p62-gels were also observed in mouse embryonic fibroblasts (MEFs) isolated from *p62-GFP* knockin mice (*p62-GFP$^{KI/+}$*), in which C-terminal GFP-tagged p62 is driven by endogenous p62 regulatory elements[28] (Supplementary Fig. S1). As p62 is a stress-inducible protein regulated by the Nuclear factor erythroid 2-related factor 2 (Nrf2)[29], a number of p62-GFP positive structures were formed in *p62-GFP$^{KI/+}$* MEFs upon exposure to sodium arsenite (As[III]) (Supplementary Fig. S1a). They were round, contained the S351- and S405-phosphorylated p62 forms and extensively colocalized with Keap1 and ubiquitin (Supplementary Fig. S1a). We confirmed similar properties of p62-structures in both mouse primary culture hepatocytes and HeLa cells after treatment with As[III] (Supplementary Fig. S1b, c), implying general properties of p62-gels. Consistent with previous report[28], we verified the increased levels of p62-GFP and p62 in *p62-GFP$^{KI/+}$* MEFs under the As[III]-treated conditions, which decreased after the removal of As[III] in a time-dependent manner (Supplementary Fig. S1d). Such decrease was abrogated by loss of *Atg7*, an essential gene for autophagy (Supplementary Fig. S1d), suggesting autophagic degradation of the p62-gels. Indeed, 19.5% of p62-gels in *p62-GFP$^{KI/+}$* MEFs colocalized with WIPI2, an isolation membrane/phagophore marker, 6 hr after removal of As[III] (Supplementary Fig. S1e). Time lapse imaging of *p62-GFP$^{KI/+}$* MEFs indicated that after removal of As[III], p62-positive structures in wild-type background gradually decreased and most of them disappeared by 12 h (Supplementary Movie S4). By contrast, in *Atg7*-knockout background, such structures remained and rather became larger due to fusion of several structures with each other (Supplementary Movie S5). Taken together, these results indicate the autophagic degradation of p62-gels.

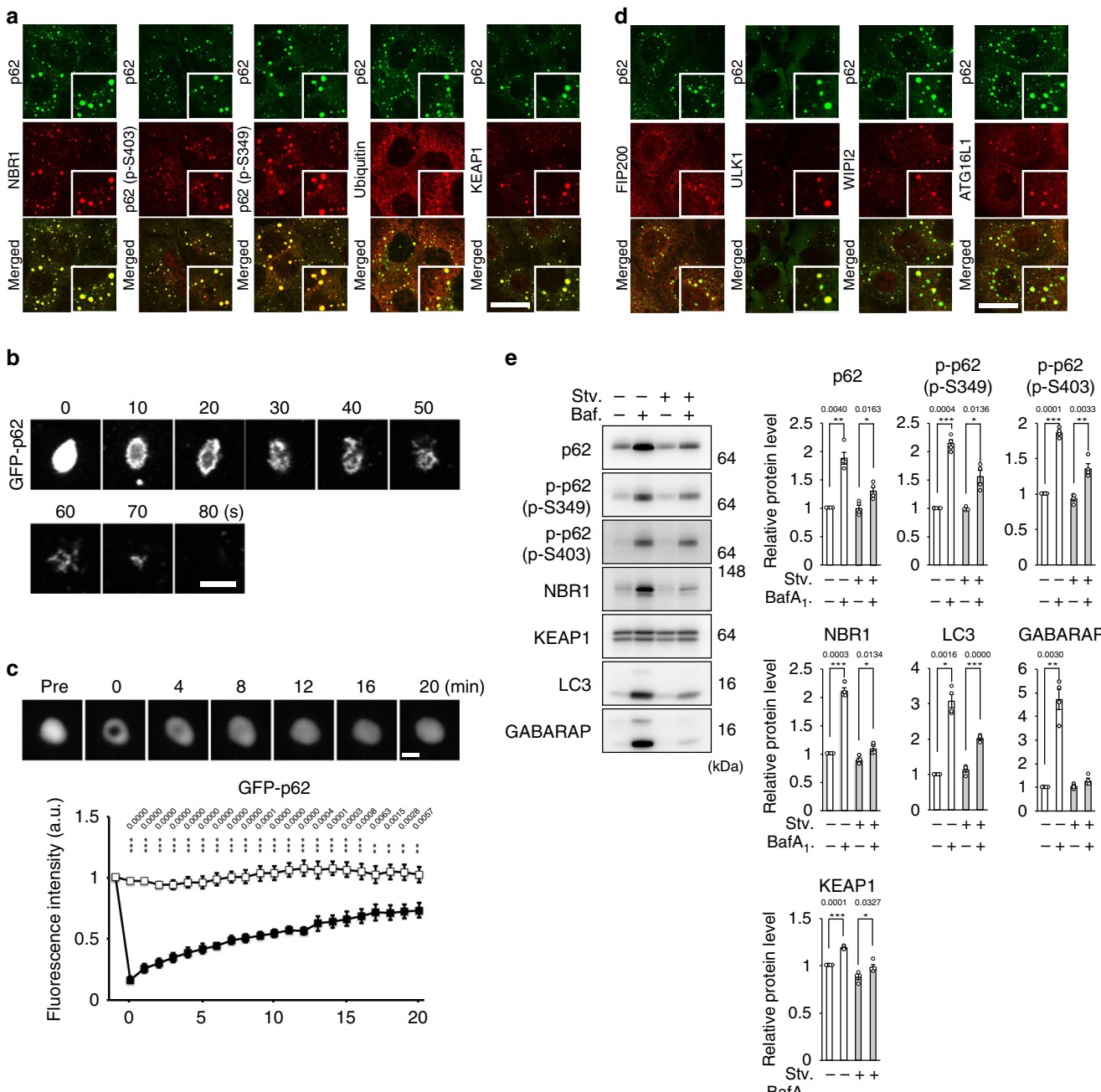

**Fig. 1 p62-gels are degraded by autophagy. a** Immunofluorescence microscopy. Huh-1 cells were immunostained with indicated antibodies. Bars: 10 μm. **b** Time lapse microscopic analysis. Huh-1 cells were transfected with GFP-p62. 24 h after the transfection, the p62-bodies labelled GFP-p62 were isolated from the cells and treated with 1,6-hexanediol. Bar: 5 μm. **c** Time lapse microscopic analysis. Huh-1 cells were infected by GFP-p62. 24 h after the infection, the p62-bodies labelled by GFP-p62 were photobleached, and then time of fluorescent recovery was measured. Data are means ± s.e of nonphotobleached ($n = 7$) and photobleached ($n = 10$) dots. **p < 0.01, and ***p < 0.001 as determined by two-sided Welch's t-test. Bar: 2 μm. Source data are provided as a Source Data file. **d** Immunofluorescence microscopy. Huh-1 cells were immunostained with indicated antibodies. Bar: 10 μm. **e** Immunoblot analysis. Huh-1 cells were cultured in regular medium in the presence or absence of 100 nM Bafilomycin A$_1$ (Baf.) for 24 h or cultured in amino acid-deprived medium (Stv.) for 4 h in presence or absence of Baf. Cell lysates were prepared and subjected to immunoblot analysis with the indicated antibodies. Data shown are representative of four separate experiments. Bar graphs indicate the quantitative densitometric analysis of the indicated proteins relative to whole proteins estimated by Ponceau-S staining ($n = 4$). Data are means ± s.e. *p < 0.05, **p < 0.01, and ***p < 0.001 as determined by two-sided Welch's t-test. Source data are provided as a Source Data file.

**Autophagosome formation on p62-gels.** Correlative light-electron microscopy (CLEM) was performed to visualize the autophagy events during clearance of p62-GFP gels in *p62-GFP$^{KI/+}$* MEFs 3 h after removal of As[III]. Notably, in electron microscopy samples, even without correlation with GFP fluorescence, p62-GFP gels could be identified by their unique morphology (Fig. 2a–c), which has also been reported before[30]. In agreement with immunofluorescence analysis with WIPI2 antibody (Supplementary Fig. S1e), the p62-GFP gels were frequently observed inside isolation membranes/phagophores and autophagosomes, which in many cases had multiple double membranes on top of each other around them (Fig. 2a–f). Three-dimensional reconstructions were

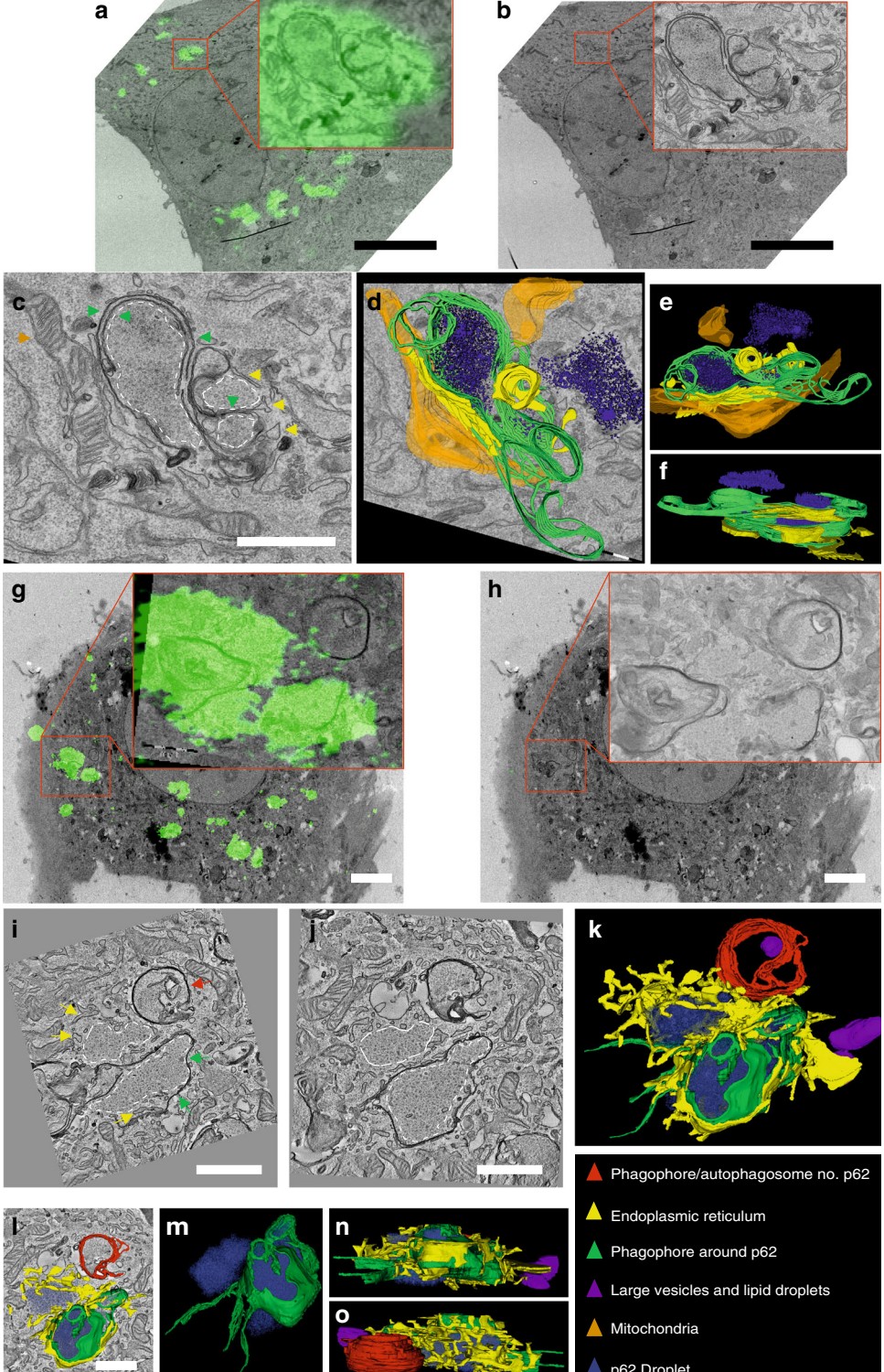

**Fig. 2 Ultrastructure and three-dimensional models of autophagic membranes around p62-gels. a–f** 80-nm TEM serial sectioning. *p62-GFP*<sup>KI/+</sup> MEFs were treated by sodium arsenite (As[III]) for 10 h and then cultured in regular medium for 3 h. **a, b** Correlation of GFP fluorescence and TEM. p62-gels, identified by their unique morphology and by correlation with the GFP fluorescence, are indicated by dotted lines in panels **c**, **i** and **j**. Bars: 10 μm. **c** The p62-gels are surrounded by single or multiple isolation membranes/phagophores or autophagosome limiting membranes (green arrows). Endoplasmic reticulum (yellow arrows) is observed next to the p62-gels and isolation membranes/phagophores or autophagosomes. Bar: 1 μm. Panels **d–f** show a three-dimensional model segmented using aligned 80-nm TEM serial sections. The color code is shown in the bottom right panel. **g, h** Correlation of GFP fluorescence and a 200-nm TEM section used for tomography. Bars: 5 μm. **i, j** Single views from the electron tomogram, showing p62-gels (dotted lines), an isolation membrane/phagophore containing p62-gel (green arrows), and an isolation membrane/phagophore or autophagosome that does not contain p62-gel (red arrow). Bars: 0.5 μm. Panels **k–o** show a three-dimensional model segmented using four aligned 200-nm electron tomograms. The color code is shown in the bottom right panel.

prepared using serial sections and electron tomography (Fig. 2c–o), and they demonstrated the unusually complex morphology of the isolation membranes/phagophores and autophagosome limiting membranes. Endoplasmic reticulum (ER) was frequently observed to surround the p62-GFP gels (Fig. 2c–f, i–o). Isolation membranes/phagophores and autophagosomes often segregated a p62-GFP gel as a whole (Fig. 2c–e). In addition, autophagosomes and isolation membranes/phagophores were frequently observed next to p62-GFP gels, but not segregating the gel (Fig. 2i–l). Approximately 50% (49 out of 99) of the phagophores/isolation membranes and autophagosomes locating next to p62-gels were enveloping p62-gel, while the rest were enveloping other cytoplasmic components. Similar to the autophagy-intact MEFs, also in $Atg7^{-/-}$; $p62\text{-}GFP^{KI/+}$ MEFs, ER was typically surrounding the p62-GFP gels (Supplementary Fig. S2b, c). In agreement with the immunoblot data (Supplementary Fig. S1d), only very few isolation membranes/phagophores and autophagosomes were observed in $Atg7^{-/-}$; $p62\text{-}GFP^{KI/+}$ MEFs, and none of them segregated p62-GFP gels (Supplementary Fig. S2e). On the basis of the morphological analysis, we concluded that there are two distinct ways of autophagosome formation on the p62-gels, either the p62-gel is engulfed by the forming autophagosome, or the autophagosome forms next to the p62-gel but does not engulf it. These two types of segregation probably correspond to selective and bulk autophagy.

**Engulfment of p62-gels by autophagosome is dependent on LC3/GABARAP interaction**. Which factor(s) decide(s) between the two distinct ways of autophagosome formation on the p62-gels? We hypothesized that the interaction of p62 with LC3 and GABARAP is critical; if p62 interacts with LC3 and GABARAP, p62-gel is engulfed by the autophagosome, if not, autophagosomes form next to the p62-gel but do not engulf it. To test our hypothesis, we initially conducted an in vitro analysis using purified recombinant proteins and giant unilamellar vesicles (GUVs). Consistent with previous reports[17], mixing of mCherry-p62 with linear tetraubiquitin (4xUb) resulted in the formation of condensates, which was not affected by LIR mutation (W338A, L341A) in p62 (Fig. 3a). We then prepared Atg8-GUVs that contain Atg8-phosphatidylethanolamine conjugates by incubating GUVs with the Atg8 conjugation system and ATP, which were previously utilized to analyze the interaction with Ape1 condensates[13]. The Atg8-GUVs were used for interaction analysis with p62-4xUb condensates since the ability to bind LIR motifs is evolutionarily conserved among Atg8 family proteins[31]. When p62-4xUb condensates were incubated with Atg8-GUVs, both dispersed p62 and 4xUb and their condensates were bound to Atg8-GUVs. Bound condensates were confirmed not to be membranous structures using GUVs with fluorescently labeled lipids (Fig. 3b). On the other hand, both of them were only scarcely bound to Atg8-GUVs when W338A, L341A mutation was introduced to p62 (Fig. 3b, c). These data indicate that p62-4xUb condensates interact with Atg8-GUVs depending on the LIR-Atg8 interaction, which is consistent with previous report[32], proposing that p62-gels could be tethered to isolation membranes via LIR-LC3/GABARAP interaction.

In the next experiments, we sought to clarify this issue in vivo. p62 LIR mutants were not used since a recent report showed that the LIR of p62 overlaps with an interacting region of FIP200, which is an upstream ATG protein[8]. Instead, we utilized a HyD-LIR-Venus construct, consisting of HyD, a short hydrophobic domain of Aplysia phosphodiesterase 4 short-form; LIR of TP53INP2; and a fluorescent protein, Venus[33]. This protein efficiently binds to LC3-II and GABARAP-II localizing on the inner membrane of autophagosomes and does not affect autophagosome formation[33] (Fig. 4a). If the interaction of p62

with LC3-II and/or GABARAP-II directs the isolation membranes/phagophores to the p62-gels, the overexpression of HyD-LIR-Venus should inhibit autophagosome sequestration of p62-gels by the competitive inhibition (Fig. 4a). We generated knockin mice that express HyD-LIR-Venus under CAG promoter in a Cre-recombinase-dependent manner ($HyD\text{-}LIR^{flox/flox}$) (Supplementary Fig. S3). $HyD\text{-}LIR^{flox/flox}$ mice were crossbred with Alb-Cre transgenic mice[34] that express the Cre recombinase under the control of the Albumin promoter to generate hepatocyte-specific HyD-LIR-Venus expressing mice ($HyD\text{-}LIR^{flox/flox}$; Alb-Cre). As expected, marked expression of HyD-LIR-Venus protein was recognized in primary hepatocytes isolated from $HyD\text{-}LIR^{flox/flox}$; Alb-Cre but not control $HyD\text{-}LIR^{flox/flox}$ mice (Fig. 4b). The expression of HyD-LIR-Venus hardly affected the conversion of LC3-I to LC3-II, rather it increased the levels of LC3-II (Fig. 4b). Treatment with BafA$_1$ increased the amount of LC3-II in both control and HyD-LIR expressing hepatocytes (Fig. 4b). Due to high expression, we did not observe any fluctuation of HyD-LIR-Venus by treatment with BafA$_1$. Immunofluorescence analysis with LC3 antibody showed colocalization of HyD-LIR-Venus with LC3-positive structures (Fig. 4c). In agreement with the biochemical analysis, the number of LC3-positive puncta in hepatocytes expressing HyD-LIR-Venus was much higher than that in control hepatocytes (Fig. 4c). Whereas the LC3-positive dots increased in number upon nutrient deprivation in control hepatocytes, such increase in the HyD-LIR-Venus expressing hepatocytes was milder (Fig. 4c), due to large number even in nutrient-rich conditions.

We observed a prominent accumulation of p62 in HyD-LIR expressing hepatocytes regardless of nutrient conditions, which was not further increased by treatment with BafA$_1$ (Fig. 4b). The treatment with BafA$_1$ slightly but significantly upregulated the amount of S351-phosphorylated p62 and its client proteins, NBR1 and Keap1 (Fig. 4b). Immunofluorescence analysis showed a prominent accumulation of p62-positive structures in hepatocytes overexpressing HyD-LIR-Venus (Fig. 4c). Although we detected p62-positive structures in control hepatocytes, their size and number were smaller than those in HyD-LIR-Venus overexpressing hepatocytes (Fig. 4c). We developed adenovirus vector expressing HyD-LIR-Venus, a tool that enabled easy expression of HyD-LIR-Venus regardless of cell types at high level. Similar to HyD-LIR-Venus expressing hepatocytes, p62-positive structures were formed by simple infection of HeLa cells with the HyD-LIR-Venus adenovirus, and they showed spherical shape (Supplementary Fig. S4). To investigate if these p62-structures exhibit gel-like or aggregate-properties, we carried out time lapse imaging. As shown in Supplementary Movie S6, the p62-structures moved through the cytoplasm and occasionally fused with each other. These results indicate that the p62-positive structures formed by the expression of HyD-LIR-Venus are liquid droplets and/or gels.

WIPI2, a core autophagy-related protein, extensively colocalized with the p62-gels in both control and HyD-LIR Venus expressing hepatocytes (Fig. 4d), implying autophagosome formation on the p62-gels. We observed signals for HyD-LIR-Venus next to or overlapping with the p62-gels (Fig. 4c). Immunoelectron microscopy (IEM) of HyD-LIR Venus expressing hepatocytes revealed that p62-gels with round or elliptical shape contained numerous vesicles and electron-dense materials (Fig. 4e). Similar structures with diameters of approximately 1 μm were often observed by conventional electron microscopy. (Fig. 4f). Autophagosome/phagophore profiles were occasionally observed in proximity of the p62-gels, but they very rarely enwrapped the gels (Fig. 4f). We have also provided the following evidence: LC3-positive puncta are colocalized with p62-structures[35], p62-structures consisting of LC3/GABARAP-

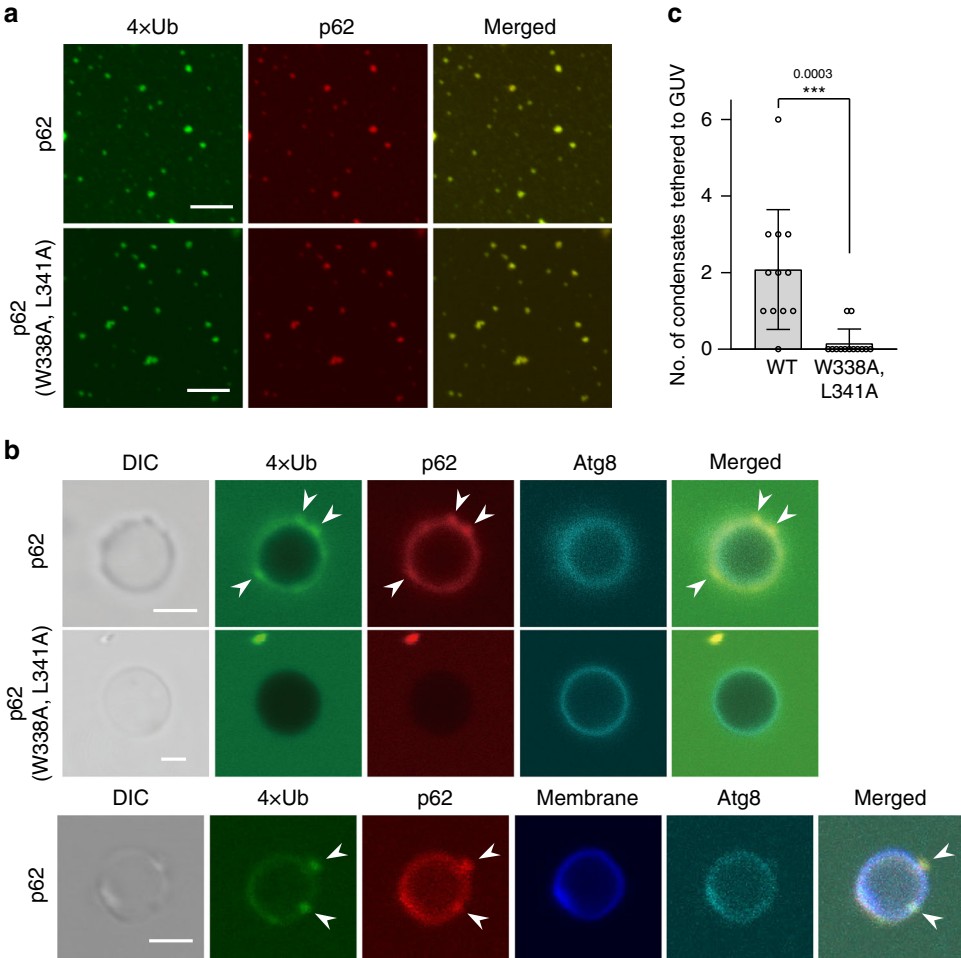

**Fig. 3 p62-4xUb condensates interact with Atg8-GUVs depending on the LIR-Atg8 interaction. a** In vitro formation of p62-4xUb condensates. 5 µM mCherry-p62, 50 µM 4xUb and 5 µM SNAP-4xUb labeled with AlexaFluor 488 were mixed and observed by fluorescence microscopy. Bars: 5 µm. **b** Interaction analysis of p62-4xUb condensates with Atg8-GUVs. p62-4xUb condensates as prepared in **a** were incubated with mKalama1-Atg8-GUVs and observed by fluorescence microscopy. DIC differential interference contrast. Arrowheads indicate p62-4xUb condensates on the Atg8-GUV. Bars: 2 µm. **c** Quantification of the experiments in **b**. GUVs were randomly selected ($n = 12$ and 13 for WT and W338A L341A, respectively) and the number of attached condensates was counted. Data are means ± s.d of wild-type and W338A L341A condensates attached with a GUV. ***$P < 0.001$ as determined by two-sided Student's $t$-test. Source data are provided as a Source Data file.

binding defective p62 are not degraded by autophagy[36], and phagophore profiles surround p62-body as shown by IEM[37]. On the basis of the published and presented evidence, we concluded that that autophagosome formation occurs on the p62-gels and that the interaction of p62 with LC3 and/or GABARAP directs the process towards the engulfment of whole p62-gels by autophagosomes.

**Impairment of LC3-interaction-dependent selective autophagy.** Despite the presence of several autophagosomes and autolysosomes in the primary hepatocytes isolated from *HyD-LIR^flox/flox*; Alb-*Cre* mice, the cells appeared to contain increased amounts of mitochondria and endoplasmic reticulum (Supplementary Fig. S5a–f). This observation implies defective mitochondria-turnover (mitophagy) and ER-turnover by autophagy (ER-phagy). Indeed, we verified prominent accumulation of mitochondrial and ER-proteins, such as Tom20, Cytochrome b, and TEX264 in the hepatocytes (Supplementary Fig. S5g). Long-lived protein degradation assay was then used to assess autophagy and revealed significantly lower degradation in the *HyD-LIR^flox/flox*; Alb-*Cre* hepatocytes compared with that in control hepatocytes (Supplementary Fig. S5h). Nutrient deprivation induced protein

degradation in hepatocytes of both genotypes, but the induction in the HyD-LIR expressing hepatocytes was significantly lower (Supplementary Fig. S5h). Protein degradation was suppressed to similar levels by the addition of lysosomal inhibitors, E64d, pepstatin A, and ammonium chloride (E64d/PepA + AC) in both genotypes (Supplementary Fig. S5h). The induced degradation in both genotypes was also inhibited by treatment with lactacystin, a proteasome inhibitor (Supplementary Fig. S5h). Considering that almost complete suppression of long-lived protein degradation has been observed in *Atg7*-deficient hepatocytes[38], these results suggest that LC3/GABARAP-interaction-dependent selective autophagy is impaired in hepatocytes of *HyD-LIR^flox/flox*; Alb-*Cre* mice, while bulk autophagy is intact.

**Nrf2 activation on p62-gels.** What is the biological significance of p62-gels? An in vitro analysis using purified recombinant proteins showed that Keap1 was recruited to p62-4xUb condensates (Fig. 5a). When the p62-4xUb-Keap1 condensates were incubated with Atg8-GUVs, they were bound to Atg8-GUVs (Fig. 5b). While GFP-p62 had an ability to form punctate structures in both wild-type and *Keap1*-deficient MEFs, structures positive for GFP-Keap1 were dispersed in *p62*-deficient MEFs

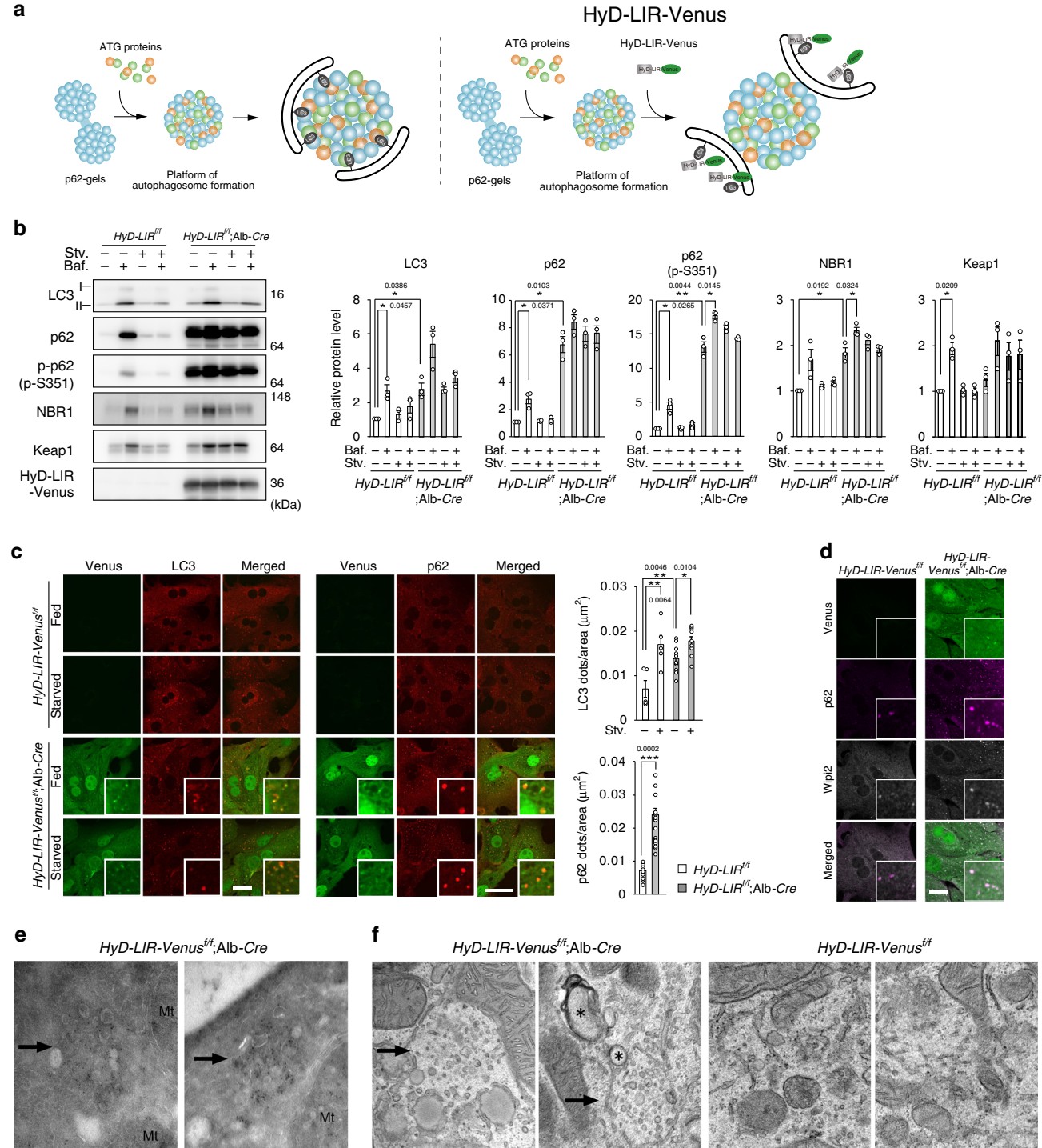

(Fig. 5c). These results suggest that Keap1 is translocated onto the p62-gels.

Keap1, an adaptor protein of cullin3-based ubiquitin ligase for Nrf2, interacts with S349-phosphorylated p62, and its binding mode is similar to that with Nrf2-ETGE motif[25,39]. We hypothesized that the p62-gel acts as a platform for Nrf2 activation through the sequestration of Keap1 onto the gel. To study this possibility, we used a chemical compound termed KMN003[40] (Fig. 6a), which is a derivative of K67. K67 is a Keap1-binding chemical compound that inhibits the interaction of Keap1 with p62[40,41]. The binding model of KMN003 with Keap1 was created based on the X-ray crystal structure of K67 in a

complex with Keap1-DC domain consisting of the double glycine repeat or kelch repeat (DGR, aa 315–598) and the C-terminal region (CTR, aa 599–624) (Protein Data Bank ID code 4ZY3)[41] using the CNS (crystallography and NMR system) energy minimization program[42]. The model showed that KMN003 is placed into the pocket at the bottom of the β-propeller structure essential for the interaction with p62 (Fig. 6b). The naphthalene ring, benzene rings and ethoxybenzene moieties of KMN003 fit into the pocket in the same manner as in the Keap1-DC–K67 complex (Fig. 6b). In addition, carboxyl group of side chain on the C-2 position of naphthalene ring interacts with R415 through a salt bridge (Fig. 6b), implying the higher inhibitory effect of

**Fig. 4 The interaction of p62 with LC3 directs autophagosomes toward the p62-gel. a** Model of the effect of HyD-LIR-Venus. **b** Immunoblot analysis. Primary hepatocytes isolated from indicated genotype mice were cultured in regular medium in the presence or absence of 100 nM Bafilomycin A$_1$ (Baf.) for 24 h or cultured in amino acid-deprived medium for 4 h (Stv.) in presence or absence of Baf. Cell lysates were prepared and subjected to immunoblot analysis with the indicated antibodies. Data shown are representative of three separate experiments. Bar graphs indicate the quantitative densitometric analysis of the indicated proteins relative to whole proteins estimated by Ponceau-S staining (n = 3). Data are means ± s.e. *P < 0.05 as determined by two-sided Welch's t-test. Source data are provided as a Source Data file. **c, d** Immunofluorescence microscopy. Primary hepatocytes described in **b** were cultured in regular medium (Fed) (**c, d**) or amino acid-deprived medium for 4 h (Starved) (**c**). The cells were immunostained with indicated antibodies. The number of LC3- and p62-positive dots in hepatocyte was determined in each genotype. Data are means ± s.e. of LC3-positive dots per area (μm$^2$) in fed (n = 5) and starved HyD-LIR$^{flox/flox}$ (n = 5), fed (n = 11) and starved HyD-LIR$^{flox/flox}$; Alb-Cre (n = 9) and p62-positive dots per area (μm$^2$) in fed (n = 12) and starved HyD-LIR$^{flox/flox}$ (n = 16). *p < 0.05, **p < 0.01, and ***p < 0.001 as determined by two-sided Welch's t-test. Bars: 20 μm. Source data are provided as a Source Data file. **e** Immunoelectron microscopy. Ultrathin-cryosections from the primary hepatocytes of indicated genotype cultured in regular medium were stained with p62 antibody (detected by colloidal gold particles of 6 nm in diameter). Arrows: p62-gel, Mt: mitochondria, Bar: 200 nm. **f** Electron microscopy. Representative electron micrographs of the primary hepatocytes of indicated genotype cultured in regular medium are shown. Arrows: p62-gel, Asterisks: autophagosome/phagophore profiles, Bars: 500 nm.

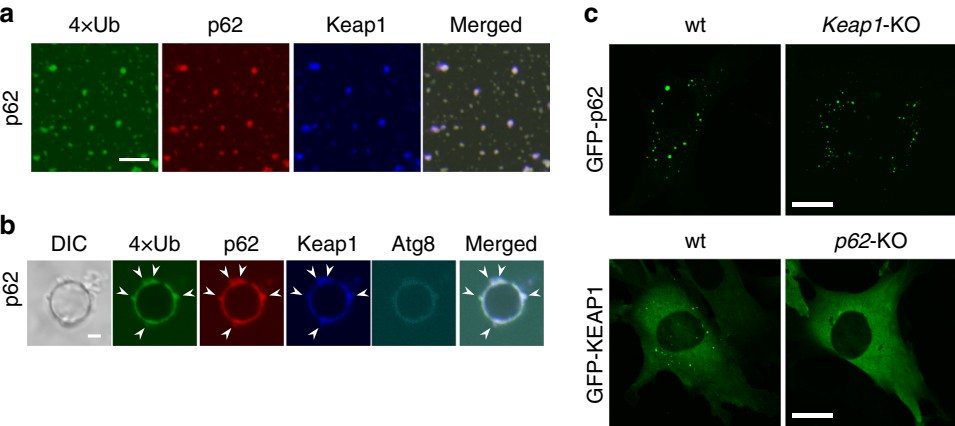

**Fig. 5 Localization of Keap1 on the p62-gels. a** In vitro formation of p62-4xUb-Keap1 condensates. 5 μM mCherry-p62, 50 μM 4xUb, 5 μM SNAP-4xUb labeled with AlexaFluor 488 and SNAP-Keap1 labelled with Alexa Fluor 647 were mixed and observed by fluorescence microscopy. Bars: 5 μm. **b** Interaction analysis of p62-4xUb-Keap1 condensates with Atg8-GUVs. p62-4xUb-Keap1 condensates as prepared in **a** were incubated with mKalama1-Atg8-GUVs and observed by fluorescence microscopy. DIC differential interference contrast. Arrowheads indicate p62-4xUb-Keap1 condensates on the Atg8-GUV. Bars: 2 μm μm. **c** Fluorescence microscopy. Wild-type, p62- or Keap1-knockout MEFs were transfected with GFP-p62 or GFP-Keap1. 24 h after the transfection, the cells were observed by confocal microscopy. Bars: 20 μm.

KMN003 to the p62-binding, compared to K67. Actually, in vitro pull-down assays revealed that KMN003 disturbed the interaction of Keap1-DC with S349-phosphorylated p62 at low concentration (Fig. 6c). Immunoprecipitation assay with Keap1 antibody showed that both p62 and S349-phosphorylated p62 were co-immunoprecipitated with Keap1, and both interactions decreased by the treatment of KMN003 in a dose-dependent manner (Fig. 6d). Treatment of Huh-1 cells with KMN003 increased the level of endogenous Keap1 (Fig. 6d), suggesting impairment of p62-mediated Keap1-turnover through autophagy[43]. Immuno-fluorescence analysis with p62 antibody revealed that the treatment of Huh-1 cells expressing GFP-Keap1 with KMN003 decreased the number of GFP-Keap1-positive p62-gels (Fig. 6e). Time lapse imaging of Huh-1 cells expressing GFP-Keap1 indicated that the GFP-Keap1-positive structures were round, moved through the cytoplasm and occasionally fused with each other similar to GFP-p62 (Supplementary Movie S7). FRAP analysis showed that after photobleaching, signal for GFP-Keap1-labelled dots recovered and that the average recovery time was 2.67 ± 0.44 min (Fig. 6f and Supplementary Movie S8), which is much faster than that of GFP-p62 (Fig. 1c and Supplementary Movie S3). We noticed that puncta positive for GFP-Keap1 often disappear upon the exposure to KMN003 (Supplementary Movie S9). While the GFP-Keap1 structures emerged in normal culture conditions (Supplementary Movie S7), we hardly observed formation of new GFP-Keap1-positive structures under

the KMN003-treated conditions (Supplementary Movie S9). These results indicate that Keap1 is translocated to the p62-gels in a p62-binding dependent fashion and that the translocation is reversible.

We next investigated the significance of Keap1 on p62-gels. When wild-type p62 or T350A-p62 mutant, which does not bind to Keap1[39], was expressed in p62-deficient Huh-1 cells, both wild-type and mutant p62 formed similar punctate structure (Fig. 7a). Keap1 colocalized with wild-type but not mutant p62 (Fig. 7a). The expression of wild-type p62 caused nuclear accumulation of Nrf2, but the T350A mutant did not (Fig. 7b). As expected, the gene expression of Nrf2-targets, such as *NAD(P)H dehydrogenase, quinone 1* (*Nqo1*), and *glutamate-cysteine ligase catalytic subunit* (*Gclc*), was markedly induced by overexpression of wild-type p62, but not the T350A mutant (Fig. 7c). We verified the increased level of Nqo1 and Gclc proteins in cells expressing wild-type p62, while such increase was not observed in cells harboring T350A-p62 (Fig. 7a). As expected, an oligomerization-defective K7A D69A mutant of p62 did not form p62-gels nor activate Nrf2 (Fig. 7a–c). Taken together, we concluded that the p62-gel acts as oxidative stress response through the sequestration of Keap1.

**The combination of Nrf2 activation with impaired autophagy causes liver injury.** Similar to the primary hepatocytes (Fig. 4), p62, phosphorylated forms of p62, NBR1 and Keap1 accumulated

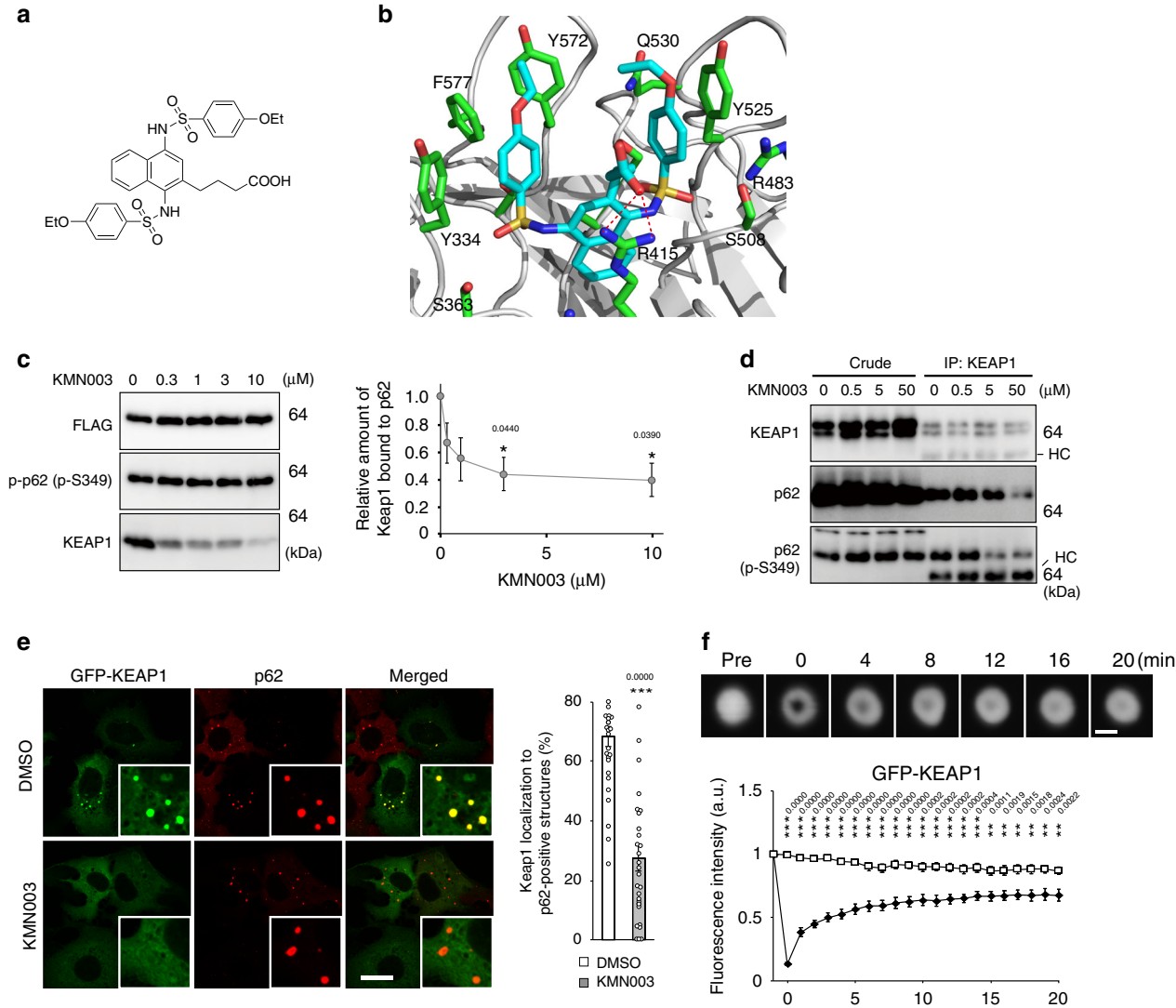

**Fig. 6 Reversible translocation of Keap1 on the p62-gels. a** Structural formulas of KMN003. (**b**) Modelling of Keap1 (white)–KMN003 (cyan). Overall structure of Keap1-DC, shown as a ribbon model. KMN003 (cyan) and some of the potential interacting residues of Keap1-DC are shown in stick representation (green). Potential hydrogen-bonds are depicted as dotted red lines. **c** In vitro pull-down assay. Recombinant Keap1-DC was allowed to form a complex with *Strep-FLAG*-tag p62, and then KMN003 was mixed with the complex at indicated concentration. Keap1-DC binding to *Strep-FLAG*-tag p62 was estimated by immunoblot analysis with Keap1 antibody. Data are representative of three independent experiments. Right graph indicates the quantitative densitometric analysis of Keap1 bound to p62 ($n = 3$). Data are means ± s.e. *$p < 0.05$ as determined by two-sided Welch's $t$-test. Source data are provided as a Source Data file. **d** Immunoprecipitation assay. Huh-1 cells were cultured in regular medium in the presence or absence of KMN003 at indicated concentration for 12 h. Cell lysates were immunoprecipitated with Keap1 antibody, and the immunoprecipitants were subjected to immunoblot analysis with the indicated antibodies. Data shown are representative of three separate experiments. HC indicates heavy chain. Source data are provided as a Source Data file. **e** Immunofluorescence microscopy. Huh-1 cells expressing GFP-Keap1 were cultured in regular medium in the presence or absence of 10 μM KMN003 for 24 h and then immunostained with p62 antibody. Data are means ± s.e. of the number of p62-positive dots with GFP-Keap1 in Huh-1 ($n = 30$). ***$P < 0.001$ as determined by two-sided Welch's $t$-test. Bars: 20 μm. Source data are provided as a Source Data file. **f** Time lapse microscopic analysis. Huh-1 cells were transfected with GFP-Keap1. 24 h after the transfection, the structures labelled by GFP-Keap1 were photobleached, and then time of fluorescent recovery was measured. Data are means ± s.e of nonphotobleached ($n = 8$) and photobleached ($n = 10$) dots. **$p < 0.01$, and ***$p < 0.001$ as determined by two-sided Welch's $t$-test. Bar: 2 μm. Source data are provided as a Source Data file.

in livers of *HyD-LIR*<sup>flox/flox</sup>; Alb-*Cre* but not in livers of control mice (Fig. 8a). The levels were lower than those in livers of *Atg7*<sup>flox/flox</sup>; Alb-*Cre* mice (Fig. 8a). As shown in Fig. 3e, both p62 and Keap1 in hepatocytes expressing HyD-LIR-Venus were degraded in lysosomes although less efficiency than in control hepatocytes, suggesting that some of p62-gels are still surrounded by autophagosomes, likely by chance. Meanwhile, autophagosome formation is severely impaired in hepatocytes lacking Atg7, and the remnant Atg7-independent autophagosomes[44,45] cannot engulf the p62-gels (Supplementary Fig. S2). Thus, it is reasonable

that levels of p62 and Keap1 in hepatocytes expressing HyD-LIR-Venus are lower than those in Atg7-deficient hepatocytes. We observed numerous punctate signals indicating p62-gels, which contained ubiquitinated proteins and S351-phosphorylated form of p62 in livers of both *HyD-LIR*<sup>flox/flox</sup>; Alb-*Cre* and *Atg7*<sup>flox/flox</sup>; Alb-*Cre* mice, but not in control livers (Fig. 8b). The p62-gels in HyD-LIR-Venus expressing hepatocytes tended to be smaller than those in *Atg7*-deficient hepatocytes, which occasionally contain large pleomorphic p62-aggregates (Fig. 8b). We speculated that Nrf2 is persistently activated in livers of *HyD-LIR*<sup>flox/flox</sup>; Alb-

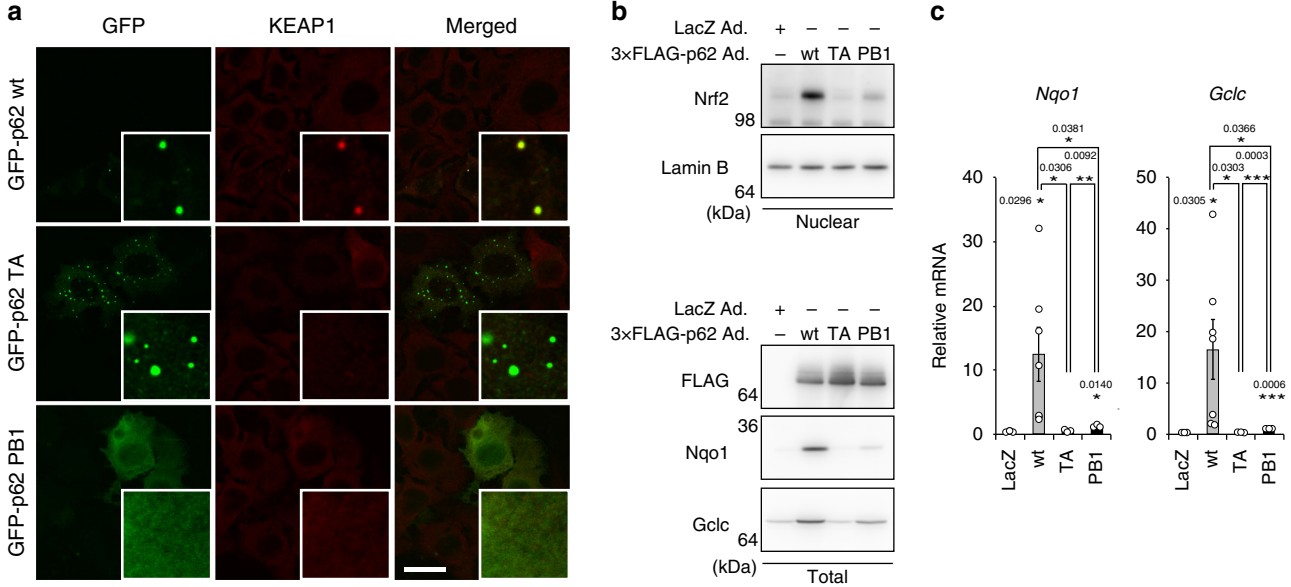

**Fig. 7 Significance of Keap1 on p62-gels. a** *p62*-deficient Huh-1 cells were transfected with wild-type p62, T350A or PB1 mutant. 24 h after the transfection, the cells were immunostained with Keap1 antibodies. Bar: 20 μm. **b** Wild-type p62, T350A or PB1 mutant was expressed in primary mouse hepatocytes using adenovirus system. 48 h after the infection, the cells were fractionated into cytosolic and nuclear fractions, which were subjected to immunoblot analysis with the indicated antibodies. Data shown are representative of three separate experiments. Source data are provided as a Source Data file. (**c**) Gene expression of Nrf2-targets. Total RNAs were prepared from primary mouse hepatocytes expressing LacZ ($n = 7$), wild-type p62 ($n = 7$), T350A ($n = 7$), or PB1 mutant ($n = 3$). Values were normalized against the amount of mRNA in LacZ-expressing hepatocytes. RT-qPCR analyses were performed as technical replicate on each biological sample. Data are means ± s.e. *$p < 0.05$, **$p < 0.01$, and ***$p < 0.001$ as determined by two-sided Welch's *t*-test. Source data are provided as a Source Data file.

*Cre* mice due to the accumulation of the p62-gels. To test this, we carried out quantitative proteomics with livers of *HyD-LIR*^flox/flox^ and *HyD-LIR*^flox/flox^; *Alb-Cre* mice and found significant upregulation (<1.5-fold) of 163 proteins including p62 in livers of *HyD-LIR*^flox/flox^; *Alb-Cre* mice (Fig. 8c and Supplementary Dataset 1). Among the 163 upregulated genes, 102 genes were found to be bound by Nrf2 in Nrf2 ChIP-seq data deposited in ENCODE and described in previous publications[46,47] (Supplementary Dataset 2). Among the Nrf2 binding sites of these 102 genes, 51 genes contained an antioxidant response element, which is a consensus sequence for Nrf2 binding (Supplementary Dataset 2). This implies Nrf2 activation in livers of *HyD-LIR*^flox/flox^; *Alb-Cre* mice. Indeed, the gene expression of Nrf2-target genes, *p62*, *Nqo1* and *glutathione S-transferase, mu 1* (*Gstm1*) was induced in livers expressing HyD-LIR-Venus, and the levels were similar to those in livers of *Atg7*^flox/flox^; *Alb-Cre* mice (Fig. 8d). We finally examined the phenotypes of *HyD-LIR*^flox/flox^; *Alb-Cre* mice, in which bulk autophagic activity is still intact. In contrast to severe liver enlargement in *Atg7*^flox/flox^; *Alb-Cre* mice, *HyD-LIR*^flox/flox^; *Alb-Cre* mice did not exhibit any hepatomegaly (Fig. 8e). Histological examination revealed that hepatocytes in HyD-LIR expressing livers were slightly larger, and possessed smaller glycogen regions compared to control liver, while *Atg7*-deficient hepatocytes had much larger eosinophilic cytoplasm than control or HyD-LIR expressing hepatocytes (Fig. 8f). The leakage of hepatic enzymes such as aspartate aminotransferase and alanine aminotransferase was evident in *Atg7*^flox/flox^; *Alb-Cre* mice, but we did not observe any significant leakage in *HyD-LIR*^flox/flox^; *Alb-Cre* mice (Fig. 8g).

## Discussion

Autophagosomes selectively surround several kinds of large protein complexes such as P-granule, stress granule, and Ape1 complex. The selective uptake of the complexes into autophagosomes is likely to be dependent on cargo liquidity. Actually, the aggregated Ape1 complex is not recognized as a selective substrate due to a failure of an adaptor protein to function on the aggregates[13]. Once stress granules are affected by proteasome inhibition, which probably converts them into gel-like structures, they become targets for autophagy[20]. Likewise, when P-granules become less dynamic gel-like structures due to heat stress, they undergo autophagic degradation[16]. When p62 binds to ubiquitin chains, it acquires liquid-like properties[17,18]. Such phase-separated droplets allow the exchange of their components, including ubiquitin and LC3, with the environment[17]. However, the properties of endogenous p62-structures remained unclear. In this study, we showed that endogenous p62-puncta are not typical liquid droplets, but gel-like structures (Fig. 1). Which factors change and/or keep the property of p62-gels? One hint is NBR1, a binding partner of p62. Since overexpression of NBR1 blocks the autophagic turnover of p62-structures[19], the quantity of NBR1 bound to p62-structures might regulate the transition from a gel-like structure to a liquid-droplet and in turn affect the direction of autophagic engulfment on the p62-gels. Since NBR1 is a stress-induced protein[19], the transition might occur under stress conditions.

The formation of the autophagosome around phase-separated p62 requires sequential and antagonistic steps. First, a complex between p62 and FIP200 is formed to initiate the selective autophagy of p62[8]. This complex is mutually exclusive to the one formed between p62 and LC3, indicating that the binding of p62 to LC3 occurs after dissociation from FIP200[8]. We observed the formation of multiple isolation membranes/phagophores on the p62-gels (Fig. 2). Such feature is distinct from starvation induced autophagy[48,49] and mitophagy[50], in which single isolation membrane/phagophore is formed. In the case of mitophagy, a ubiquitin-binding adaptor protein NDP52 that is localized on damaged and then ubiquitinated mitochondria interacts with FIP200, leading to autophagosome formation around the mitochondria[10]. Since NDP52 is unlikely to form liquid droplets, the

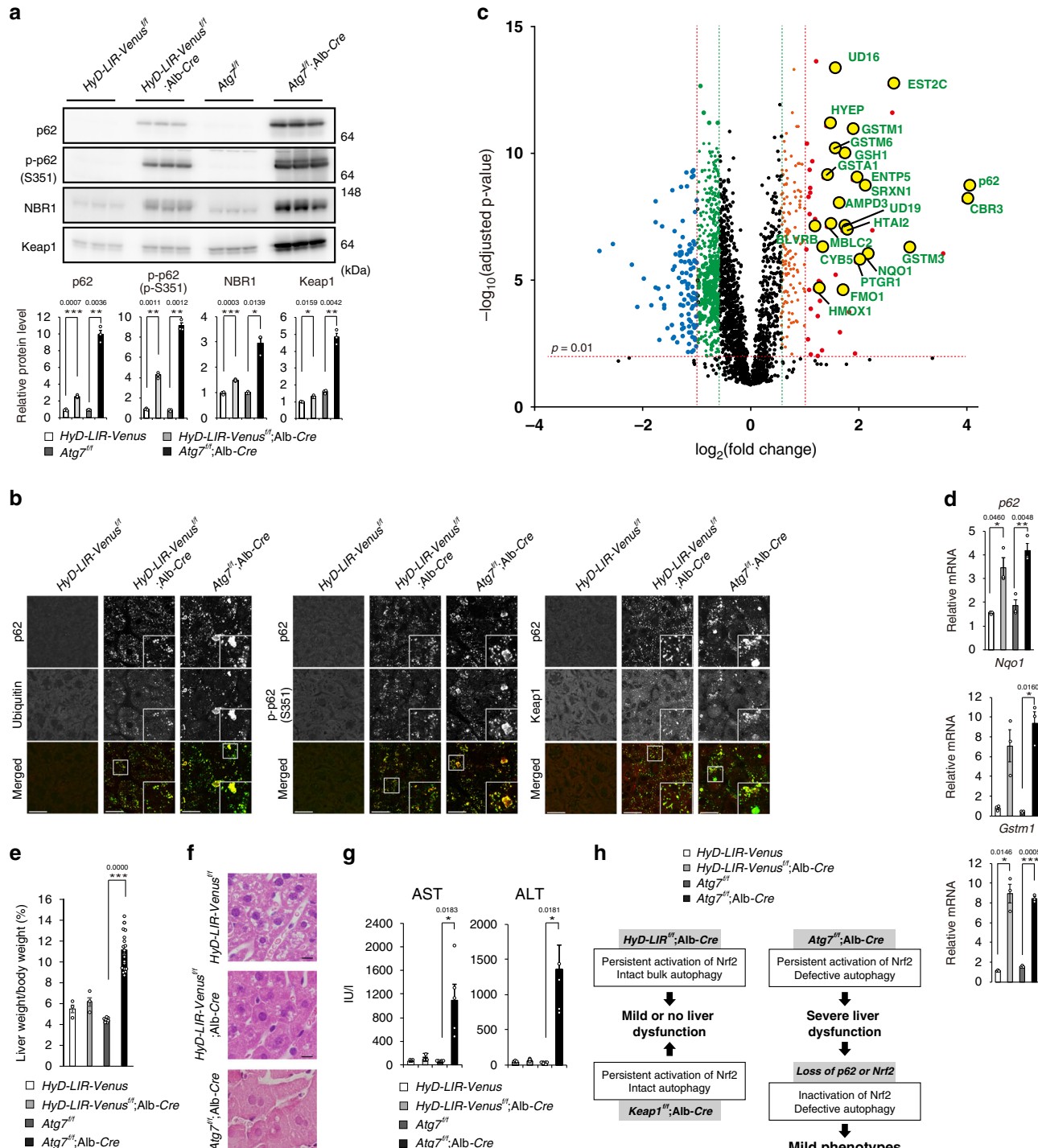

stoichiometry between NDP52 and FIP200 is likely to be lower compared with that between p62 and FIP200. Thus, it is plausible to suggest that the number of FIP200 on the cargo defines the number of isolation membranes/phagophores forming around the cargo. One p62-gel may interact with FIP200 molecules on multiple sites, which drives the formation of a number of isolation membranes/phagophores.

What is the role of LC3 in selective autophagy? It has been thought for a long time that the interaction of LC3 with selective substrates is indispensable to tether cargoes to elongating isolation membranes/phagophores. Considering selective autophagy in *Saccharomyces cerevisiae*[51] as well as recent findings showing that ubiquitin-binding adaptors interact with FIP200 in

mammalian cells[8–10], isolation membrane/phagophore should be formed along the surface of selective substrates. While the blockade of the interaction of LC3 with selective substrates by overexpression of HyD-LIR did not affect autophagosome formation next to the p62-gels, isolation of p62-gels into the forming autophagosomes was almost completely cancelled (Fig. 4). Thus, the LC3-binding likely acts as driving force to direct isolation membranes/phagophore toward engulfment of the p62-gels. Similarly, Atg19, an adaptor protein for Ape1, and lipidated Atg8 are necessary and sufficient for selective sequestration of Ape1 droplets by membranes[13]. In sharp contrast, loss of LC3- and GABARAP-families in HeLa cells does not affect selective engulfment of damaged mitochondria by autophagosomes[52]. We

**Fig. 8 Physiological significance of p62-turnover.** in vivo **a** Immunoblot analysis with homogenates from livers of three 5-week-old *HyD-LIR^flox/flox* and *HyD-LIR^flox/flox*; Alb-*Cre* mice. Data shown are representative of three separate experiments. Bar graphs indicate the quantitative densitometric analysis of the indicated proteins relative to whole proteins estimated by Ponceau-S staining. Data are means ± s.e. *$p < 0.05$, **$p < 0.01$, and ***$p < 0.001$ as determined by two-sided Welch's *t*-test. Source data are provided as a Source Data file. **b** Immunohistofluorescence microscopy. Paraffin sections of mouse livers described in **a** were double-immunostained with p62 and ubiquitin, S351-phosphorylated p62 or Keap1 antibodies. Bars: 20 µm. **c** Proteomics analysis with liver homogenates of mice described in **a**. Nrf2-target genes are indicated in yellow color. **d** Gene expression of Nrf2-targets. Total RNAs were prepared from mouse livers of 5-week-old *HyD-LIR^flox/flox* (n = 3) and *HyD-LIR^flox/flox*; Alb-*Cre* mice (n = 3). Values were normalized against the amount of mRNA in *HyD-LIR^flox/flox* livers. RT-qPCR analyses were performed as technical replicate on each biological sample. Data are means ± s.e. *$p < 0.05$, **$p < 0.01$, and ***$p < 0.001$ as determined by two-sided Welch's *t*-test. Source data are provided as a Source Data file. **e** Liver weight (% per body weight) of 5-week-old *HyD-LIR^flox/flox* (n = 4), *HyD-LIR^flox/flox*; Alb-*Cre* (n = 4), *Atg7^flox/flox* (n = 4), and *Atg7^flox/flox*; Alb-*Cre* (n = 4) mice. Data are means ± s.e. ***$p < 0.001$ as determined by two-sided Welch's *t*-test. Source data are provided as a Source Data file. **f** Hematoxylin and eosin staining of liver paraffin sections described in **e**. Bars: 50 µm. **g** Liver function tests of the mice used in 5-week-old *HyD-LIR^flox/flox* (n = 4), *HyD-LIR^flox/flox*; Alb-*Cre* (n = 4), *Atg7^flox/flox* (n = 4), and *Atg7^flox/flox*; Alb-*Cre* (n = 4) mice. The serum levels of aspartate aminotransferase (AST) and alanine aminotransferase (ALT) were measured. IU/L, international units/liter. Data are means ± s.e. *$p < 0.05$ as determined by two-sided Welch's *t*-test. Source data are provided as a Source Data file. **g** Schematic diagram of liver pathology.

observed slight but significant accumulation of mitochondria and prominent accumulation of ER in hepatocytes expressing HyD-LIR (Supplementary Fig. S5). The importance of LC3-binding to substrates in selective autophagy may be dependent on substrates and cell types.

What is a biological role of the p62-gels? When the activity as an autophagy adaptor for ubiquitinated proteins is not required, p62 is held inactive through homodimerization of its ubiquitin-associated (UBA) domain, which prevents it from interacting with ubiquitin. Phosphorylation events drive the liberation of the UBA domain from dimeric inactivation; in particular, phosphorylation of serine 407 of p62 by ULK1 has been shown to facilitate the transition from dimer to monomer[28]. This modification is followed by phosphorylation of p62 on serine 403, either by ULK1, casein kinase 2 (CK2) or TANK-binding kinase 1 (TBK1), which enhances its binding to ubiquitin chains[24,53,54]. Once p62 binds to ubiquitin chains, the liquid–liquid phase separation occurs[17,18]. Besides a role of p62 in the condensation of ubiquitinated proteins, we propose that the p62-gel plays a central role in oxidative stress response, namely Nrf2 activation. Keap1 was translocated onto the p62-gels, and its blockade suppressed the activation of Nrf2 (Figs. 5–7). The phosphorylation of p62 at Ser349, which is induced under selective autophagy conditions, enhances the interaction of p62 with Keap1[25]. It is worth noting that this phosphorylation is also involved in the interaction of p62 with FIP200[8]. These findings raise the possibility that once Ser349 of p62 is phosphorylated, selective autophagy and Nrf2 activation are concomitantly activated. Moreover, since the translocation of Keap1 onto the p62-gels was reversible (Fig. 6), the phosphorylation and/or de-phosphorylation of p62-Ser349 may participate in the regulation of both selective autophagy of p62-gels and oxidative stress response.

Due to marked accumulation of p62 and its phosphorylated forms in liver-specific autophagy-deficient mice (e.g. hepatocyte-specific *Atg5*- and *Atg7*-knockout mice), Nrf2 is constitutively activated, which is believed to cause liver pathogenesis such as hepatomegaly, liver injury and development of adenoma[12,35,39,55,56]. However, these results were derived from phenotypic analyses of *p62*- or *Nrf2*-knockout mice under autophagy-deficient background. It was not unveiled whether Nrf2 activation, which is generally thought to be cytoprotective, is really cytotoxic or not. Utilizing *HyD-LIR^flox/flox*; Alb-*Cre* mice, in which the turnover of p62-gels was compromised, we found that while liver Nrf2 is activated at similar extent to liver-specific *Atg7*-knockout liver, the mice did not show any pathological signs (Fig. 8). These results indicate that simple Nrf2 activation does not exert any cytotoxic effect on the liver and that the combination of Nrf2 activation with impaired autophagy causes liver injury. Consistently, liver-specific *Keap1*-knockout

mice, in which Nrf2 is constantly activated without affecting autophagy, exhibit slight liver pathology[57]. Although we do not exclude the possibility that more severe liver phenotypes in *Atg7*-knockout mice are derived from both defective autophagy and LC3-associated phagocytosis (LAP), it was reported recently that LAP-incompetent but autophagy-intact mice show no liver pathologies[58]. In conclusion, when both anabolism (Nrf2 activation) and catabolism (autophagy) are concomitantly deregulated, severe liver injury occurs (Fig. 8h).

In summary, our data indicate that endogenous p62 is present as gel-like structures, the p62-gels serve as platforms for autophagosome formation and for Nrf2 activation, and that the p62-mediated Nrf2 activation itself is not cytotoxic.

## Methods

**Cell culture.** Huh-1 (JCRB0199), MEFs, and HeLa (ATCC CCL-2) cells were grown in Dulbecco's modified Eagle medium (DMEM) containing 10% fetal bovine serum (FBS), 5 U/ml penicillin, and 50 µg/ml streptomycin. For overexpression experiments, Huh-1 cells were transfected with Lipofectamine 3000 (Thermo Fisher Scientific, Waltham, MA, USA). *p62-GFP^KI/+* and *Atg7^−/−*; *p62-GFP^KI/+* MEFs were used in this study. Huh-1 and HeLa cells were authenticated by STR profile. All cell lines were tested for mycoplasma contamination.

**Mice.** For the generation of *Rosa26-loxP-stop-loxP-HyD-LIR-Venus* mice (*HyD-LIR-Venus* mice), a genomic fragment that contained the ROSA locus was isolated from C57BL/6 mouse genomic BAC clone from an RP23 mouse genomic BAC library (Advanced Geno TEchs Co). The targeting vector had a splice acceptor (SA), two loxP sequences that were inserted before the repeated SV40 poly-adenylation signal and after the "stop" sequence that contained the terminator of the yeast His3 gene and SV40 polyadenylation signal. A 1.7-kb fragment of the FRT-PGK-gb2-neo-FRT-loxP cassette (Gene Bridges) was inserted after the repeated SV40 polyadenylation signal. After the second loxP sequence, the targeting vector contained the coding sequence of the DTR, simian HBEGF followed by IRES and tdTomato sequences. The targeting vectors were electroporated into mouse RENKA ES cells, selected with G418 (250 µg/ml; Invitrogen, San Diego, CA), and then screened for homologous recombinants by Southern blot analysis. Southern blot analysis in *HyD-LIR-Venus* mice was performed by digesting genomic DNA with *Nde*I (Takara Bio, Inc., Shiga, Japan) or *Eco*RV (Takara Bio, Inc.), followed by hybridization to detect wild-type 12.7-kb and flox 9.7-kb bands or wild-type 11.5-kb and flox 19.3-kb bands. Mice were housed in specific pathogen-free facilities, and the Ethics Review Committee for Animal Experimentation of Juntendo University approved the experimental protocol. We have complied with all relevant ethical regulations.

**RT-qPCR (real-time quantitative reverse transcriptase PCR).** Using the Transcriptor First-Strand cDNA Synthesis Kit (Roche Applied Science, Indianapolis, IN, USA), cDNA was synthesized from 1 µg of total RNA. RT-qPCR was performed using the LightCycler® 480 Probes Master mix (Roche Applied Science) on a LightCycler® 480 with LightCycler® 480 Software version 1.5.1 (Roche Applied Science). Signals from human and mouse samples were normalized against *GAPDH* (glyceraldehyde-3-phosphate dehydrogenase) and *Gusb* (ß-glucuronidase) mRNA, respectively. The sequences of primers used for gene expression analysis in either mouse livers or human cell lines are provided in Supplementary Table S1.

**Immunoblot analysis**. Livers were homogenized in 0.25 M sucrose, 10 mM 2-[4-(2-hydroxyethyl)-1-piperazinyl]ethanesulfonic acid (HEPES) (pH 7.4), and 1 mM dithiothreitol (DTT). Cells were lysed in ice-cold TNE buffer (50 mM Tris-HCl, pH 7.5, 150 mM NaCl, 1 mM EDTA) containing 1% SDS and protease inhibitors. Nuclear and cytoplasmic fractions from cultured cells were prepared using the NE-PER Nuclear and Cytoplasmic Extraction Reagents (Thermo Fisher Scientific). For immunoprecipitation analysis, Huh-1 cells were grown on six-well plates and treated with KMN003 for 12 h at the indicated concentrations. The cells were lysed in 300 μl of IP buffer (20 mM Tris-HCl [pH 7.5], 150 mM NaCl, 1 mM EDTA, 1% NP40, 1%TX-100) containing Protease inhibitor cocktail (Roche), and the lysates were then centrifuged at 20,000 × g for 10 min at 4 °C to remove debris. In the next step, 200 μl of IP buffer, 0.5 μl of anti-Keap1 antibody (10503-2-AP, PGI) and 10 μl of Protein G Sepharose 4 Fast Flow (Cytiva) were added to the 200 μl of lysate, and the mixture was mixed under constant rotation for 3 h at 4 °C. The immunoprecipitates were washed three times with ice-cold IP buffer. The complex was boiled for 5 min in SDS sample buffer in the presence of β-Mercaptoethanol to elute proteins. For pull-down assay, HEK293T cells were transfected with One-Strep-FLAG-tagged p62 (OSF-p62). Twenty-four hour after the transfection, the cell lysate was subjected to StepTactin Sepharose (Cytiva), and then purified Keap1-DC (321-609aa) was added to the StepTactin Sepharose-conjugated OSF-p62. The complex consisting of Keap1-DC and OSF-p62 was treated in presence or absence of the KMN003 in pull-down assay buffer (20 mM Tris-HCl [pH 7.5], 150 mM NaCl, 0.5% NP40, 0.5 mM TCEP). The pulled-down protein complexes were washed with pull-down assay buffer and then boiled for 5 min in SDS sample buffer in the presence of β-Mercaptoethanol to elute proteins. Samples were subjected to SDS-PAGE, and transferred to a polyvinylidene difluoride membrane thereafter (Merck, IPVH00010). Antibodies against p62 (GP62-C, Progen Biotechnik GmbH, Heidelberg, Germany; 1:1000), S403-phosphorylated p62 (GTX128171, GeneTex, CA, USA; 1:1000), NBR1 (4BR; Santa Cruz Biotechnology, Dallas, TX USA; 1:1000), LC3B (#2775, Cell Signaling Technology; 1:500), KEAP1 (10503-2-AP, proteintech, Rosemont, USA; 1:1000), Nqo1 (ab34173; Abcam, Cambridge, UK; 1:2000), Gclc (ab41463, Abcam; 1:500), GSTm1 (GSTM12-S, ALPHA DIAGNOSTIC INTERNATIONAL, San Antonio, USA; 1:1000), and GFP (594, Medical and Biological Laboratories, Nagoya, Japan; 1:1000) were purchased from the indicated suppliers. Anti-S349-phosphorylated p62 polyclonal antibody (1:1000) was raised in rabbits by using the peptide Cys+KEVDP(pS)TGELQSL as an antigen. Blots were incubated with horseradish peroxidase-conjugated goat anti-mouse IgG (H + L) (Jackson ImmunoResearch Laboratories, Inc., 115-035-166;), goat anti-rabbit IgG (H + L) (Jackson ImmunoResearch Laboratories, Inc., 111-035-144; 1:10000) or goat anti-guinea pig IgG (H + L) antibody (Jackson ImmunoResearch Laboratories, Inc., 106-035-003; 1:10000), and visualized by chemiluminescence. Band density was measured using the software Multi Gauge V3.2 (FUJIFILM Corporation, Tokyo, Japan). Uncropped and unprocessed scans of the all immunoblots were supplied in the Source Data file.

**Immunofluorescence microscopy**. Cells grown on coverslips were fixed in 4% paraformaldehyde in PBS for 10 min, permeabilized with 0.1% Triton X-100 or 50 ng/ml Digitonin in PBS for 5 min, blocked with 0.1% (w/v) gelatin (Sigma–Aldrich) in PBS for 45 min, and then incubated overnight with primary antibodies diluted 1:200 in gelatin/PBS. After washing, cells were incubated with Goat anti-Guinea pig IgG (H + L) Cross-Adsorbed Secondary Antibody, Alexa Fluor 488 (A11073, Thermo Fisher Scientific), Goat anti-Mouse IgG (H + L) Highly Cross-Adsorbed Secondary Antibody, Alexa Fluor 647 (A21236, Thermo Fisher Scientific) or Goat anti-Rabbit IgG (H + L) Highly Cross-Adsorbed Secondary Antibody, Alexa Fluor 647 (A21245, Thermo Fisher Scientific) at a dilution ratio of 1:1000 for 60 min. For treatment of 1,6-Hexandiol, GFP-p62 was transfected in Huh-1 cells. twenty-four hour after transfection, the cells were lysed in ice-cold lysis buffer (50 mM Tris-HCl, pH 7.4, 150 mM NaCl, 1 mM EDTA) containing 0.2% Triton X-100 and 65 U/mL RNasin® Ribonuclease Inhibitors (Promega). The lysates were further passed through 27-gauge needle 20 times and incubated for 20 min at 4 °C. The resulting lysates were centrifuged at 200 × g for 5 min. The supernatant was mixed with 10 mM MgSO4, 1 mM CaCl2 and 4 U/mLRQ1 RNase Free DNase (Promega) and incubated for 30 min at RT. The mixture was centrifuged at 10,000 × g for 7 min. The pellet containing GFP-p62 labelled droplets was mixed with 1,6-Hexandiol (final concentration 10%). Cells and droplets were imaged using a confocal laser-scanning microscope FV1000 with FV10-ASW 04.01 or FV3000 with FV31S-SW 2.4.1.198 (Olympus) with a UPlanSApo ×60 NA 1.40 oil objective lens. To measure the fluorescence recovery after photobleaching (FRAP), cells were grown in 35 mm glass base dishes (Iwaki, Japan). p62-bodies were bleached for 3 sec using a laser intensity of 70% at 480 nm, and then the recovery was recorded for the indicated time. After image acquisition, contrast and brightness were adjusted using Photoshop 2021v25.0 (Adobe).

**Correlative light and electron microscopy (CLEM)**. Detailed CLEM method was recently published[59]. Briefly, *p62-GFP^{KI/+}* MEFs were cultured on coverslips etched with 150-μm grids (Matsunami Glass Ind., Ltd.). The MEFs were treated by sodium arsenite (As[III]) for 10 hr and then cultured in regular medium for 3 h. They were fixed with 2% PA–0.5% glutaraldehyde (GA) in 0.1 M PB (pH 7.4), and phase-contrast and fluorescence images were obtained using a confocal microscope (FV1000; Olympus). The cells were fixed again with 2% PA and 2% GA in 0.1 M

PB (pH 7.4), processed using the reduced-osmium method, and embedded in Epon812. Areas containing cells of interest were trimmed, cut as serial 70-nm or 200-nm sections, and observed with an electron microscope (EM; JEM1400 Plus; JEOL). Alignment of light microscopy and EM images was performed using TrakEM2 software version 1.0a 2012 -07-04[60] (https://imagej.net/TrakEM2).

**Tomography**. Electron tomography was performed on 200-nm serial sections using 15-nm gold particles deposited on both surfaces of the grids as fiducial markers. JEM1400 Plus operated at 120 kV was used to collect dual-axis tilt series from −65° to + 65° with 1.5° steps using SerialEM version 3.3.1[61]. Tomograms were generated using the IMOD software package version 4.7.3[62]. The tomograms where segmented in Microscopy Image Browser (version 2.651/25.04.2020)[63] and then exported to 3D Slicer software (https://www.slicer.org/)[64] in order refine the models and make the final images. ImageJ (the core package in Fiji) version ImageJ 1.53c (https://imagej.net/Fiji)[65] was used for general image handling. Fig. 2 and Supplementary Fig. S2 were created using Corel draw version 19.0.0.328 (https://www.coreldraw.com/en/).

**Electron microscopy (EM) and immuno-EM for primary hepatocytes**. For conventional EM, primary hepatocytes were fixed in 0.1 M PB containing 2% paraformaldehyde and 2% glutaraldehyde, followed by a process of the reduced-osmium method[66], embedding in Epon812, sectioning, and staining with uranyl acetate and lead citrate. For immune-EM, the cells were fixed in 0.1 M PB containing 4% paraformaldehyde and 4% sucrose, and then frozen after infusion with 2.3 M sucrose and 20% polyvinylpyrrolidone[67]. Ultrathin sections were mounted on Formvar carbon-coated nickel grids, blocked with 1% bovine serum albumin (BSA) in PBS, incubated with anti-p62 (GP62-C, Progen) antibody, and then incubated with colloidal gold (6 nm)–conjugated goat anti-guinea pig antibody (Jackson ImmunoResearch laboratories, Inc). The sections were observed with an electron microscope (EM; JEM1400EX; JEOL)

**Immunohistofluorescence microscopy**. Mouse livers were quickly excised, cut into small pieces, and then fixed by immersing in 4% paraformaldehyde-4% sucrose in 0.1 M phosphate buffer, pH 7.4 (PB). After rinsing, they were embedded in paraffin or OCT compound for hematoxylin-eosin staining and immunostaining. Paraffin- and cryosections were prepared and processed for antigen retrieval for 20 min at 98 °C using a microwave processor (MI-77, AZUMAYA, Japan) in 1% immunosaver (Nissin EM, Japan). Sections were blocked and incubated for 2–3 days at 4 °C with the following first antibodies: guinea pig polyclonal antibody against p62 (GP62-C, Progen), rabbit polyclonal antibody against phosphorylated p62[25] or ubiquitin (Z0458, Dako), and mouse monoclonal antibody against Keap1 (Proteintech Group). They were further incubated Alexa647-conjugated donkey anti-guinea pig IgG (706-605-148, Jackson ImmunoResearch laboratories, Inc., West Grove, PA), and Alexa594-conjugated donkey anti-rabbit IgG (711-585-152, Jackson ImmunoResearch laboratories, Inc.) or Alexa594-conjugated donkey anti-mouse IgG (715-585-150, Jackson ImmunoResearch laboratories, Inc.). HE-images were acquired with a microscope (BX51, Olympus, Japan) equipped with a cooled CCD camera system (DP-71, Olympus), and immunofluorescence images were taken by a laser-scanning confocal microscope (FV1000, Olympus) with 60x objective lens (PlanApo N 60x, NA 1.42 oil). After image acquisition, contrast and brightness were adjusted using Photoshop CS6 (Adobe).

**Plasmids for recombinant protein expression in bacteria**. The pGEX6P-1 vector (GE Healthcare) was selected for production of N-terminal glutathione S-transferase (GST) fusion proteins in E. coli cells. For construction of pGEX6P-mCherry-p62, the gene encoding mCherry-p62 was amplified by PCR and inserted into the downstream region of HRV 3C protease recognition site. The gene encoding linear tetraubiquitin (4xUb) was purchased from GENEWIZ and inserted into the downstream region of HRV 3C protease recognition site of the pGEX6P-1 vector. For construction of pGEX6P-SNAP-4xUb, the gene coding SNAP-tag was inserted into the upstream region of 4xUb of pGEX6P-4xUb. For construction of pGEX6P-SNAP-Keap1, the gene coding Keap1 with an N-terminal SNAP-tag was inserted into the downstream region of the pGEX6P-1 vector. All gene insertions were performed by NEBuilder HiFi DNA Assembly (New England Biolabs). Mutations to generate the indicated amino acid substitutions were introduced by PCR-mediated site-directed mutagenesis. All constructs were sequenced to confirm accuracy of cloning.

**Protein expression and purification**. E. coli strain BL21(DE3) cells were used for expression of all recombinant proteins. Protein expression was induced by 0.1 mM isopropyl β-D-thiogalactopyranoside (IPTG) at 16–18 °C overnight. The E. coli cells were harvested by centrifugation and lysed by sonication in Buffer A (50 mM HEPES-NaOH pH 8.0, 500 mM NaCl) supplemented with 1 mM Tris(2-carboxyethyl)phosphine (TCEP), 1 mM phenylmethylsulfonyl fluoride, and 1x protease inhibitor cocktail (EDTA free) (Nacalai tesque). For the E. coli cells expressing GST-SNAP-Keap1, DNase I (Takara) and RNase A (Nacalai tesque) were added to Buffer A. After removal of insoluble cell debris by centrifugation, the supernatant was applied to GST accept resin (Nacalai tesque). After washing the resin with Buffer A, HRV 3C protease with a GST-tag was added to the resin. After the on-

column cleavage for ~16 h at 4 °C, the cleaved proteins were collected and concentrated to at least 50 µM for mCherry-p62 WT, mCherry-p62 W338A L341A, and SNAP-Keap1 or at least 500 µM for SNAP-4xUb and 4xUb in Buffer A supplemented with 10% glycerol using Vivaspin 500 (GE Healthcare). SNAP-4xUb and SNAP-Keap1 were labelled with SNAP-Surface Alexa Fluor 488 and SNAP-Surface Alexa Fluor 647 (New England Biolabs), respectively. mKalama1-Atg8K26P and Atg8 conjugation enzymes (Atg3, Atg7, Atg16, and Atg5-Atg12 conjugate) were prepared as previously reported[13]. Purified proteins were stored at −80 °C until use.

**Preparation Atg8-GUVs.** GUVs were prepared as previously reported[13]. Briefly, 100 µL of a 1 mM phospholipid mixture consisting of 1-palmitoyl-2-oleoyl-sn-glycero-3-phosphocholine (POPC), 1-palmitoyl-2-oleoyl-sn-glycero-3-phosphoethanolamine (POPE), L-α-phosphatidylinositol (Liver, PI) (all from Avanti Polar Lipids) at a molar ratio of 5:4:1 in chloroform was prepared in a 5-mL glass vial and dried under a gentle stream of nitrogen to produce a thin, homogeneous lipid film. For GUVs labelled with Cy5, 0.01 mM 1,2-dioleoyl-sn-glycero-3-phosphoethanolamine-N-(Cyanine 5) (Cy5-DOPE) (Avanti Polar Lipids) was added to 1 mM phospholipid mixture. The glass vial was put in a vacuum desiccator overnight. After thin lipid film was prehydrated with 20 µL of MilliQ water at 60 °C for 7 min, 500 µL of Buffer B (20 mM HEPES pH 7.5, 150 mM NaCl, 0.1 M sucrose) was added and incubated for 2–3 hr at 60 °C. The solution containing GUVs was transferred to 1.5-mL microcentrifuge tube and centrifuged at 13,000 × g for 30 min at 25 °C. After removal of precipitates, the GUV solution was diluted with Buffer C (20 mM HEPES pH 7.5, 150 mM NaCl, 0.1 M glucose). For the lipidation of Atg8 on the GUV membrane (Atg8-GUV), a 10 µL mixture of 1 µM mKalama1-Atg8K26P, 0.5 µM Atg3, 0.5 µM Atg7, 0.25 µM Atg12-Atg5-Atg16 in Buffer C supplemented with 1 mM ATP and 1 mM MgCl$_2$ was added to a 10 µL GUV solution and incubated for 1 h at ~23 °C.

**In vitro observations of p62-4xUb condensates formation and interactions between Atg8-GUV and p62-4xUb condensates.** Fluorescence observation was performed on glass-bottom dishes (MatTek) coated with 0.3% (w/v) bovine serum albumin in Buffer C using an FV3000RS confocal laser-scanning microscope (Olympus). 405, 488, 561, and 640-nm lasers were used for excitation of mKalama1, Alexa Fluor 488, mCherry and Alexa Fluor 647 or Cy5, respectively. To observe p62-4xUb condensates, 5 µM mCherry-p62 or mCherry-p62 W338A L341A was mixed with 50 µM 4xUb and 5 µM labelled SNAP-4xUb dissolved in 20 µL of Buffer D (20 mM HEPES-NaOH pH 7.5, 150 mM NaCl, 1 mM TCEP). To observe p62-4xUb-Keap1 condensates, 5 µM mCherry-p62 was mixed with 50 µM 4xUb, 5 µM labelled SNAP-4xUb and 5 µM labelled SNAP-Keap1 dissolved in 20 µL of Buffer D. To observe the interactions between Atg8-GUV (or Atg8-GUV labelled with Cy5) and p62-4xUb condensates, 5 µM mCherry-p62 or mCherry-p62 W338A L341A was mixed with 20 µL Atg8-GUV solution containing 50 µM 4xUb and 5 µM labelled SNAP-4xUb. To observe the interactions between Atg8-GUV and p62-4xUb-Keap1 condensates, 5 µM mCherry-p62 was mixed with 20 µL Atg8-GUV solution containing 50 µM 4xUb, 5 µM labelled SNAP-4xUb and 5 µM labelled SNAP-Keap1. Each mixed solution was incubated for 1 h at ~23 °C before observation.

**Mass spectrometry analysis.** Mass spectrometry analyses were performed on a TripleTOF 5600 mass spectrometer coupled to Ekspert nanoLC 415 system with Ekspert cHipLC (Sciex, Framingham, MA). The samples were injected onto a Trap column of 200 µm × 0.5 mm ChromXP C18-CL, 120 Å pore size, 3 µm diameter particles and 75 µm × 150 mm ChromXP C18-CL analysis column (Sciex). Samples were run with a 100 min gradient from 2–40% solvent B (solvent A: 0.1% formic acid, solvent B: 0.1% formic acid, 80% acetonitrile) at a flow rate of 300 nl/min. The analysis results were searched against the UniProt database (release 2019_11) using ProteinPilot 5.0.2. software (Sciex). The library contained 2619 protein groups with the cutoff of 1% false discovery rate. Similar LC gradient and mass spectrometer settings as for the IDA acquisition described above were used, but the mass spectrometer was operated in SWATH mode. The precursor isolation windows overlap by 1 m/z and cover a range of 400–1250 *m/z*. The variable isolation windows were determined with SWATH variable window calculator tool version 1.0 (Sciex). The MS2 spectra were recorded with an accumulation time of 40 ms and cover 100–1600 *m/z*. PeakView 2.2 (Sciex) was used to analyze the SWATH data. Proteins with one to five peptides were used for quantitative analysis. The sum of peptide peak area was used for protein quantification.

Unpaired student's *t*-tests were performed to compare the SWATH results for the HyD-LIR-Venus expressing liver groups and control groups. FDR corrections for the *P* values were carried out with Storey's method[68,69]. The up- and downregulated proteins with the adjusted *P* values less than 0.01 were used for the following the DAVID enrichment analysis[70]. The DAVID analysis was performed to obtain detailed information on biological functions and pathways that significantly enriched up- and downregulated proteins in SWATH analysis.

**Analysis of ChIP-seq dataset for NRF2 binding.** NRF2 ChIP-seq data in MEL and CH12 cells were obtained from ENCODE database via Integrative Genomics Viewer (IGV 2.3.65). Those in diethylmaleate-treated Hepa1c1c7 cells and *Keap1*-

deficient mouse bone-marrow-derived macrophages were obtained from previous publications[46,47]. If the submit of NRF2 binding peaks were contained within the gene body and flanking intergenic regions in at least one of the four NRF2 ChIP-seq data sets, the gene was considered to have NRF2 binding. The NRF2 binding regions were searched for the antioxidant response element (GCnnn$^G/_C$TCA$^T/_C$), which is a consensus sequence of NRF2 binding.

**Protein degradation assay.** Hepatocytes were plated at $5 \times 10^4$ cells/well in collagen-coated 24-well plates and cultured in Williams'E medium with 10% FCS (Williams E/10 % FCS) for 24 hr. Cells were incubated with Williams E/10 % FCS containing [$^{14}$C]-leucine (0.5 µCi/ml) for 24 h to label long-lived proteins. Cells were washed with Williams E/10 % FCS containing 2 mM unlabelled leucine and incubated with the medium for 2 h to allow degradation of short-lived proteins and minimize the incorporation of labeled leucine, released by proteolysis into protein. The cells were then washed with PBS, and incubated at 37 °C with Krebs–Ringer bicarbonate (KRB) medium and Williams E/10% FCS in the presence or absence of protease inhibitors [5 mM monomethylamine (MA), 10 µg/mL E64d and pepstatin (E/P) or 5 µM epoxomicin]. After 4 h, aliquots of the medium were taken and a one-tenth volume of 100% trichloroacetic acid was added to each aliquot. The mixtures were centrifuged at 12,000 × g for 5 min, and the acid-soluble radioactivity was determined using a liquid scintillation counter. At the end of the experiment, the cultures were washed twice with PBS, and 1 ml of cold trichloroacetic acid was added to fix the cell proteins. The fixed cell monolayers were washed with trichloroacetic acid and dissolved in 1 ml of 1 N NaOH at 37 °C. Radioactivity in an aliquot of 1 N NaOH was determined by liquid scintillation counting. The percent protein degradation was calculated according to the published procedure.

**Purification of p62-gels.** To purify p62-gels from Huh-1 cells, we modified and used a method for purification of P-body in cells that was developed by elsewhere[71]. Briefly, GFP-tagged p62 was expressed in Huh-1 cells. Twenty-four hour after transfection, the cells were suspended with ice-cold lysis buffer (50 mM Tris-HCl, pH 7.4, 150 mM NaCl, 1 mM EDTA) containing 65 U/mL RNasin® Ribonuclease Inhibitors (Promega) and EDTA-free Protease inhibitor cocktail (Roche) and then lysed by passing 20 times through a 27Gx3/4. The lysates were incubated on ice for 20 min and then centrifuged at 200 × g for 5 min at 4 °C to delete nuclei. The resultant supernatants were supplemented with 10 mM MgSO$_4$ and 1 mM CaCl$_2$ and treated with 4 U/mL of RQ1 DNase (Promega) for 30 min at RT. Thereafter, the mixtures were spun at 10,000 × g for 7 min at 4 °C, and pellets were resuspended into 10 mL of lysis buffer with 80 Units of RNaseOut (Promega). From this fraction, p62-bodies were sorted on BD FACSAria™ II Cell Sorter with BD FACSDiva software v8.0 (BD BioSciences). Particles were detected according to their Forward-scattered light (FSC) and their green fluorescence using the 488 nm excitation laser and the 526/52 band pass filter.

**Statistics and reproducibility.** Values, including those displayed in the graphs, are means ± s.e. Statistical analysis was performed by the unpaired *t*-test (Welch test) with Microsoft Excel 2011 (Microsoft). A *P* value less than 0.05 was considered to indicate statistical significance. Each experiment was repeated independently at least three times with similar results.

**Reporting summary.** Further information on research design is available in the Nature Research Reporting Summary linked to this article.

## Data availability

The accession numbers are PXD019492 for ProteomeXchange and JPST000830 or jPOST. All figures and movies are available in figshare (https://doi.org/10.6084/m9.figshare.13218746). Source data are provided with this paper.

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

## Acknowledgements

We thank all members of Komatsu's lab. and Sánchez-Martín Pablo for their helpful comments and discussion. S. Kageyama is supported by a Grant-in-Aid for Scientific Research (C) (20K06549). Y.-S.S. is supported by a Grant-in-Aid for Scientific Research (C) (19K15043). D.N. is supported by a Grant-in-Aid for Early-Career Scientists (19K16344). N.N.N. is supported by a Grant-in-Aid for Scientific Research on Innovative Areas (19H05707). M.K. is supported by a Grant-in-Aid for Scientific Research on Innovative Areas (19H05706), a Grant-in-Aid for Scientific Research (B) (18H02611), the Japan Society for the Promotion of Science (an A3 foresight program), and the Takeda Science Foundation (to M.K.). This work was supported by JSPS KAKENHI Grant Number JP 16H06276 (AdAMS). E.L.E. and S.R.G. were supported by the Academy of Finland and Magnus Ehrnrooth Foundation. The Electron Microscopy Laboratory at Institute of Biomedicine, University of Turku, in thanked for technical help and availability of instruments.

## Author contributions

M.K. designed and directed the study. E-L.E., S.W., N.N.N., and M.K. wrote the manuscript. S. Kageyama, Y-S.S., Y.I., and T.U. performed the cell biological experiments. S.R.G. and E-L.E. carried out CLEM and tomography. N.T. and S.W. performed histological and microscopic analyses with genetic modified mice. D.N. and N.N.N. performed GUV experiments. M.A. and K.S. generated genetic modified mice. S. Kazuno and Y.M. conducted proteomic analyses with genetically modified mice. H.M. searched Nrf2-targets among upregulated proteins in genetically modified mice. J-A.L. provided HyD-LIR probe. T.O. developed and provided KMN003. S.O. and N.O. conducted statistic analysis of proteomics data. All authors discussed the results and commented on the manuscript.

## Competing interests

The authors declare no competing interests.
