## [Peer Review File · Nature Communications]

REVIEWER COMMENTS

Reviewer #1 (Remarks to the Author):

In this manuscript the authors study the role of p62 condensates in the nucleation of autophagosomes as well as in the degradation of Keap1 and antioxidative stress response. While they perhaps expectedly found that p62 condensates template the nucleation and expansion of isolation membranes, they developed a very potent tool (HyD-LIR) to disrupt the interaction of LIR motif containing proteins including p62 with LC3/GABARAP proteins. Using this tool, they could uncouple the induction of autophagosome formation by p62 condensates from the attachment of the LC3 decorated expanding isolation membrane to the condensates. In addition, they could show using the HyD-LIR tool and an inhibitor, which disrupts the p62 – Keap1 interaction that the induction of the oxidative stress response alone is not responsible for the hepatomegaly.

This is a very strong manuscript and the study will be interesting for scientists working in the fields of phase separation, oxidative stress response and autophagy. I have only a few relatively minor comments.

1. The authors should describe how the GFP-p62 structures were purified from Huh-1 cells (lines 111 – 223).
2. The observation that p62 is relatively immobile in condensates/structures was already made by Sun et al. and Zaffagnini et al. (Refs 16, 17). These papers should be cited in line 119.
3. The authors use CLEM to image p62-GFP gels in cells and state that they have a unique morphology (lines 147 – 149). They should compare their results with those of Jakobi, 2020, Nature Comms. Do the authors also detect filamentous structures?
4. Lines 155 – 156: It is stated that connections between the isolation membranes and the ER are observed. It seems that the resolution is not high enough to distinguish connections from areas where the membranes are very close. Either the authors corroborate this statement or they should tone it down.
5. Figure 2b, c: The authors should elaborate on how they distinguish GUV bound condensates from small GUVs that are tethered to larger GUVs via p62-Atg8 interactions.
6. Lines 186 – 188: The observation that p62 structures can be directly tethered to Atg8/LC3 positive GUVs was already made by Wurzer et al., 2015, eLife. This reference should be included.
7. Figure 4e: The authors should explain what HC stands for. I assume “heavy chain”.
8. Figure 5a: Why is the effect of HyD-LIR expression on the Keap1 levels so small compared to the Atg7 deficient cells? Could the authors please elaborate on this?
9. The authors should briefly discuss that Atg7 is also required for the conjugation of LC3/GABARAP proteins in non-autophagic pathways such as LAP. Thus, the fact that HyD-LIR expressing mice do not phenocopy Atg7 deficient mice might be related to these non-autophagic roles of Atg7. The authors may want to discuss this aspect.

Reviewer #2 (Remarks to the Author):

Comments to the Authors:

The present study is aimed at elucidating the mechanism and the functional relevance of the

degradation of p62 droplets. The authors first confirmed a gel-like property of p62 droplets that contained p62, ubiquitin and Keap1. They next show that multiple autophagosomes formed at the p62 droplets and that this formation depended on the interaction of p62 with LC3 and GABARAP. They also showed that translocation of Keap1 to the p62 droplets required the interaction of these two proteins and was necessary for Keap1 activation. Finally, using a mouse model with defective interaction of LC3 and GABARAP with p62 droplets in hepatocytes, the authors further demonstrate that failure to remove p62 droplets by autophagy leads to activation of Nrf2 and a mild liver phenotype compared to hepatocyte-specific autophagy deficient mice.

The study provides novel insights into the property of p62-structures being gel-like as opposed to liquid droplets and shows the necessity for autophagy to clear p62-gels through interaction with LC3II. However, the study also raises questions about the biological significance of the p62-gels.

Major comments:

1. The study uses only one cell system (cancer cell line) that expresses high level of p62 to perform most of the studies. Therefore, it is not clear if most of the observations in this cell type will apply to other cell types that do not express high level of p62. How generalizable are these observations to normal cells?
2. The authors observed that some p62-gels are engulfed by autophagosomes but some are not such as in Figure 2. In those that did not get engulfed, is the interaction with LC3II impaired? Is there something else that can decide whether p62 droplets get engulfed or not?
3. How is it that autophagosomes can still form in the absence of LC3II in Atg7^{-/-}; p62-GFPKI/+ MEFs (Figure S2)? And if some of them still form, why are they not interacting with p62-gels?
4. The authors suggest that overexpression of HyD-LIR did not affect autophagy but reduces selective autophagy as evidenced by reduced mitochondria and ER turnover and defective protein degradation and a massive accumulation of p62 structures. First, are the p62-structures in HyD-LIR hepatocyte gel like or are they aggregates? Second, p62 accumulation is also known to inhibit proteasome function, was that measured?
5. The other surprising finding of the current study is the fact that overexpression of HyD-LIR impaired selective autophagy, which affected several organelles in hepatocytes, yet no obvious liver defects were observed when compared to liver-specific Atg7^{-/-} deficient livers despite an increase in Nrf2 activation. This let the reviewer to think that aberrant accumulation of selective autophagy cargo is harmless to the cells? Are the mitochondria normal? Do they respire normally? Is there any superoxide leakage?
6. Please include the controls (HyD-LIR^{flox/flox}) for Figures 3g and 3h.
7. Figure 3f (left), need to include p62 and Figure 3 (right) need to include LC3. It is also suggested to give different letters for these two images.

Minor comments:

Page 9, line 282, change "increased" to decrease.
Abstract, line 43, change "distinct to" to "distinct from"

Reviewer #3 (Remarks to the Author):

In this manuscript, authors shown P62 body is gel form by phase separation, moreover, they shown formation of autophagosome on p62 droplet. Finally, they shown recruitment of Keap1 and activation of Nrf-2 on p62 droplet and the combination of Nrf2-activation with autophagy impairment cause liver damage.

The formation of p62 droplet by phase separation have been reported recently, however, the physiological significance of phase separation of p62 is not clear, by shown p62 droplet is the site of Nrf2 activation and selective autophagosome formation, authors provide important conceptual advance to the field of selective autophagy, it is a comprehensive study and for most part, data are compelling. I have a few suggestions to further improve this study.

- 1) Authors need to shown the recruitment of Keap1 into p62 droplet in vitro.
- 2) Authors conclude Nrf2 activation on p62 gel by disruption of p62/Keap1 interaction by chemical inhibitor and using p62 mutant which can't bind to Keap1, which are nice, however, this reviewer argue these data is open to alternative explanation and does not provide a direct evidence for Nrf2 activation on p62 gels. Authors should try p62 mutants which can't form droplet.
- 3) Does Atg1 or other autophagy machinery localized on p62 droplet? If so, it will greatly support author's conclusion that autophagosome form on p62 gel.

Li Yu

Reviewer #4 (Remarks to the Author):

In this manuscript, the authors show that p62-positive puncta within the cells are gel-like structures. The p62-gels are degraded by selective autophagy through their engulfment by autophagosome, which depends on p62 interaction with LC3/GABARAP proteins. Ultimately, the p62 gels serve as a platform for activation of Nrf2 through the sequestration of Keap1 into the p62 gels.

Major comments

- The authors nicely show that in Huh-1, an hepatocellular carcinoma cell line expressing high level of p62, the p62 structures are gels. To rule out the possibility that the existence of p62 gels depends on p62 expression levels and/or it is cell type-specific, it is important show that p62 also organizes in gel in non-carcinogen (hepatic) cells expressing low or moderate levels of p62.
- In the text (line 147) as well as in the Material and Methods, the authors refer to CLEM. None of the figures (Figure 2 and Figure S2), however, depicting EM data show the corresponding fluorescence images used for the correlation and the undoubtable identification of the structures of interest. The CLEM data have to be added to the paper since they will help to determine which structures have been solved at the ultrastructural level.
- In line with the previous comment, in Figure 2 the authors claim that the EM micrographs show either p62 gel, phagophore containing p62 gels, or autophagosome that do not contain p62 gel. Without any labelling (immuno-EM and/or CLEM), based on which criteria the authors can establish which one of these 3 structures does contain or not p62?
- How the authors can distinguish the morphology of a phagophore from an autophagosome especially using 3D reconstruction on serial sections that do not resolve entirely the complete structures? In particular, the authors cannot conclusively state that an autophagosome is an autophagosome if its entire structure has not been resolved as the opening, which will classify it as a phagophore, could be present in the part of the has not been resolved/imaged. The authors should consider to use focus-ion beam-scanning electron microscopy (FIB-SEM) in order to resolve the complete 3D-volume of their structures and therefore undoubtedly distinguish autophagosomes from phagophores.
- "On the basis of the morphological analysis, we concluded that there are two distinct ways of

autophagosome formation on the p62-gels, either the p62-gel is engulfed by the forming autophagosome, or the autophagosome forms next to the p62-gel but does not engulf it". EM are steady-state images, therefore the p62 gel not engulfed may just represent an earlier event, prior the subsequent engulfment. The 2 types of segregation probably correspond to selective and bulk autophagy, this latter occur at basal level in numerous cell types. This has to indicated in the text.

- Lines 149-151 and lines 157-161, without quantification, these observations are highly speculative.

- Line 150 "They were frequently observed inside isolation membranes/phagophores and autophagosomes, which in many cases had multiple double membranes around them". Only the EM micrograph on Figure 2a show a clear "multiple double membrane", while the rest of the micrographs does not. In addition, the enwrapping of an autophagic cargo by multiple double membranes have already been shown for mitophagy by one of the authors. This has to be mentioned in the text.

- To confirm their conclusion that the engulfment of p62 gel by autophagosome is dependant of the interaction between p62 and LC3/GABARAP, the authors should show by IEM or CLEM that LC3 or GABARAP are present onto the forming phagophore surrounding the p62 gel.

Minor comments

- It would be highly beneficial for the reader that the authors precisely define and then are consistent with the terminology such as body, liquid droplet and gel structure. In the current version of the manuscript, these terms are not always properly used and/or easily mixed.

- In the FRAP experiment (line 115 and Figure 1c), the authors observed that the fluorescence recovery time of the GFP-p62-positive structures is slower than the one generally obtained for liquid droplets, leading to the fact that then p62 structures are gels. Does the recovery time of the GFP-p62 gels correspond to the ones observed for others gels described in the literature?

- Line 79, "... called LIR/GIM to interact with autophagosome-localizing ATG8 ...", use a single nomenclature, LIR/GIR or AIM/GIM

- Line 537, "... the cells were fixed in 0.1 M PB containing 4% paraformaldehyde and 4% sucrose...", it seems that "sucrose" should be replaced by glutaraldehyde

Referee's comments (italized)

Referee #1

General comments:

5 *In this manuscript the authors study the role of p62 condensates in the nucleation of autophagosomes as well as in the degradation of Keap1 and antioxidative stress response. While they perhaps expectedly found that p62 condensates template the nucleation and expansion of isolation membranes, they developed a very potent tool (HyD-LIR) to disrupt the interaction of LIR motif containing proteins including p62 with LC3/GABARAP proteins. Using this tool, they could uncouple the induction of autophagosome formation by p62 condensates*
10 *from the attachment of the LC3 decorated expanding isolation membrane to the condensates. In addition, they could show using the HyD-LIR tool and an inhibitor, which disrupts the p62 – Keap1 interaction that the induction of the oxidative stress response alone is not responsible for the hepatomegaly.*

15 *This is a very strong manuscript and the study will be interesting for scientists working in the fields of phase separation, oxidative stress response and autophagy. I have only a few relatively minor comments.*

Reply:

20 We would like to thank the Referee for the positive reception of our manuscript and the thoughtful advice regarding how to improve it.

Major comments:

Comment-1:

25 *The authors should describe how the GFP-p62 structures were purified from Huh-1 cells (lines 111 – 223).*

Reply-1:

30 We apologize for lacking description of this procedure in the original manuscript. We described the details in the materials and methods section of the revised manuscript (Page 23, lines 722-735 in the revised manuscript).

Comment-2:

35 *The observation that p62 is relatively immobile in condensates/structures was already made by Sun et al. and Zaffagnini et al. (Refs 16, 17). These papers should be cited in line 119.*

Reply-2:

We cited both papers (Page 4, line 119 in the revised manuscript).

Comment-3:

40 *The authors use CLEM to image p62-GFP gels in cells and state that they have a unique morphology (lines 147 – 149). They should compare their results with those of Jakobi, 2020, Nature Comms. Do the authors also detect filamentous structures?*

Reply-3:

45 The fine structure of the p62-GFP gels we see in conventional plastic sections is very similar to the morphology described in Jakobi et al. (Nat Commun 440, 2020). We also observe short filament-like structures in the gels. Jakobi et al. 2020 is now cited in the manuscript (Page 5, line 156 in the revised manuscript).

50 Comment-4:

Lines 155 – 156: It is stated that connections between the isolation membranes and the ER are observed. It seems that the resolution is not high enough to distinguish connections from areas where the membranes are very close. Either the authors corroborate this statement or they should tone it down.

55

Reply-4:

We agree that the resolution in the images is not very good for detection of membrane connections. Therefore, we have omitted this statement and the corresponding image.

60 Comment-5:

Figure 2b, c: The authors should elaborate on how they distinguish GUV bound condensates from small GUVs that are tethered to larger GUVs via p62-Atg8 interactions.

Reply-5:

65 We studied the interaction of p62-4xUb condensates with Atg8-GUVs containing fluorescently-labeled lipids and showed that the bound puncta did not contain fluorescently-labeled lipids (Figure 3b in the revised manuscript), indicating that the puncta on Atg8-GUVs are condensates rather than small GUVs (Page 6, lines 192-193 in the revised manuscript).

70 Comment-6:

Lines 186 – 188: The observation that p62 structures can be directly tethered to Atg8/LC3 positive GUVs was already made by Wurzer et al., 2015, eLife. This reference should be

included.

75 Reply-6:

We cited the paper (Page 7, line 197 in the revised manuscript).

Comment-7:

Figure 4e: The authors should explain what HC stands for. I assume “heavy chain”.

80

Reply-7:

Yes. HC means “heavy chain”. In the Figure legend, we described it (legend of Figure 4g in the revised manuscript).

85 Comment-8:

Figure 5a: Why is the effect of HyD-LIR expression on the Keap1 levels so small compared to the Atg7 deficient cells? Could the authors please elaborate on this?

Reply-8:

90 As shown in Figure 3e, both p62 and Keap1 in hepatocytes expressing HyD-LIR were degraded in lysosomes though less efficiency than in control cells, suggesting that some of p62-bodies are still surrounded by autophagosomes, most likely by chance. Meanwhile, autophagosome formation is severely impaired in hepatocytes lacking Atg7, and the remnant autophagosomes cannot engulf the p62-gels due to defective LC3- and GABARAP-localization on the
95 autophagosomes. Thus, it is reasonable that levels of p62 and Keap1 in hepatocytes expressing HyD-LIR are lower than in *Atg7*-deficient hepatocytes. We describe above-mentioned things in the revised manuscript (Page 11, lines 334-341 in the revised manuscript).

Comment-9:

100 *The authors should briefly discuss that Atg7 is also required for the conjugation of LC3/GABARAP proteins in non-autophagic pathways such as LAP. Thus, the fact that HyD-LIR expressing mice do not phenocopy Atg7 deficient mice might be related to these non-autophagic roles of Atg7. The authors may want to discuss this aspect.*

105 Reply-9:

Thank you for the comment. Though we do not exclude the possibility that more severe liver phenotypes in *Atg7*-knockout mice are derived from defects in both autophagy and LC3-associated phagocytosis (LAP), it was reported recently that LAP-incompetent but autophagy-

intact mice show no liver pathologies (Rai S, Autophagy, 15, 599, 2019). We mentioned this in
110 the discussion section of the revised manuscript (Page 15, lines 449-452 in the discussion
section of the revised manuscript).

Referee #2

115 General comments:

*The present study is aimed at elucidating the mechanism and the functional relevance of the
degradation of p62 droplets. The authors first confirmed a gel-like property of p62 droplets that
contained p62, ubiquitin and Keap1. They next show that multiple autophagosomes formed at
the p62 droplets and that this formation depended on the interaction of p62 with LC3 and
120 GABARAP. They also showed that translocation of Keap1 to the p62 droplets required the
interaction of these two proteins and was necessary for Keap1 activation. Finally, using a
mouse model with defective interaction of LC3 and GABARAP with p62 droplets in hepatocytes,
the authors further demonstrate that failure to remove p62 droplets by autophagy leads to
activation of Nrf2 and a mild liver phenotype compared to hepatocyte-specific autophagy
125 deficient mice.*

*The study provides novel insights into the property of p62-structures being gel-like as opposed
to liquid droplets and shows the necessity for autophagy to clear p62-gels through interaction
with LC3II. However, the study also raises questions about the biological significance of the
p62-gels.*

130

Reply:

We would like to thank the Referee for the useful suggestions on how to improve our
manuscript.

135 Specific comments:

Comment-1:

*The study uses only one cell system (cancer cell line) that expresses high level of p62 to perform
most of the studies. Therefore, it is not clear if most of the observations in this cell type will
apply to other cell types that do not express high level of p62. How generalizable are these
140 observations to normal cells?*

Reply-1:

We have analyzed the p62-gels in Huh-1 cells, but also in normal cells, mouse embryonic
fibroblasts (MEFs) (Fig. 2 and Supplementary Figure S1a in the original manuscript). In

145 addition to those cells, we investigated the p62-gels in mouse primary culture hepatocytes and
another cancer cell line, HeLa cells. After treatment with arsenite, p62-gels were formed in all
these normal and cancer cells. Similar to the p62-gels observed in Huh-1 and MEFs, they were
round and positive for phosphorylated forms of p62, Keap1, and ubiquitin. Thus, we concluded
that the properties of p62 which we have observed Huh-1 and MEFs are general. We presented
150 the results (Supplementary Fig. S1b and c of the revised manuscript) and described the above-
mentioned data (Page 5, lines 135-137 in the revised manuscript).

Comment-2:

*The authors observed that some p62-gels are engulfed by autophagosomes but some are not
155 such as in Figure 2. In those that did not get engulfed, is the interaction with LC3II impaired?
Is there something else that can decide whether p62 droplets get engulfed or not?*

Reply-2:

Since the expression of NBR1 blocks the autophagic turnover of p62 (EMBO Rep 21, e48902,
160 2020), the decision whether the p62-gels get engulfed or not might be dependent on the quantity
of NBR1 on the gels. Considering that the selective uptake of large protein complexes into
autophagosomes is dependent on cargo liquidity (Mol Cell 77, 1163, 2020, Cell 174, 1492,
2018 and Mol Cell 70, 906, 2018), NBR1 might control the liquidity of the p62-gels and in turn
the direction of autophagosomes on the p62-gels. We discussed this possibility (Page 13, lines
165 381-385 in the discussion section of the revised manuscript).

In the EM images, we are looking at one time point. The cells do eventually clear
almost all p62-gels as demonstrated by Western blotting (Supplementary Figure S1d), but this is
accomplished by autophagic degradation of a small proportion of gels at a time. Thus, it is
likely that none of the gels will be left non-engulfed in the end.

170

Comment-3:

*How is it that autophagosomes can still form in the absence of LC3II in Atg7-/-; p62-GFPKI/+
MEFs (Figure S2)? And if some of them still form, why are they not interacting with p62-gels?*

175 Reply-3:

We observed very few autophagosomes in the Atg7-knockout MEFs. Those that we observed,
may be due to the Atg5 and Atg7-independent autophagy described in literature (Nature, 461,
654, 2009). Since conversion of LC3-I and GABARAP-I to LC3-II and GABARAP-II is totally
dependent on ATG5, ATG7 and ATG3, the autophagosomes in Atg7-deficient MEFs do not
180 recognize the p62-gels. Though less compared with that in wild-type cells, fusion between

autophagosome and lysosome is observed in *ATG3*-deficient cells (Science 2016, 354, 1036). We cited both papers (Page 11, line 339 in the revised manuscript).

Comment-4:

185 *The authors suggest that overexpression of HyD-LIR did not affect autophagy but reduces selective autophagy as evidenced by reduced mitochondria and ER turnover and defective protein degradation and a massive accumulation of p62 structures. First, are the p62-structures in HyD-LIR hepatocyte gel like or are they aggregates? Second, p62 accumulation is also known to inhibit proteasome function, was that measured?*

190

Reply-4:

To address this comment, we developed a tool that we can use to easily express HyD-LIR-Venus at high level, regardless of cell types, an adenovirus vector expressing HyD-LIR-Venus. Similar to HyD-LIR-Venus expressing hepatocytes, p62-positive structures were formed by simple infection of HeLa cells with the HyD-LIR-Venus adenovirus, and they were spherical in shape (Supplementary Figure S4 of the revised manuscript). To investigate if these p62-structures exhibit gel-like or aggregate-like properties, we carried out time lapse imaging. As shown in Supplementary Movie 6, the p62-structures moved through the cytoplasm and occasionally fused with each other, matching with the criteria of the liquid-droplets and/or gels.

200 We also measured the proteasome activity in liver of *HyD-LIR-Venus^{flax/flax}; Alb-Cre* mice and compared it with that in control liver. The 26S and 20S proteasome activities in mutant liver were comparable to those in control liver (Figures for the reviewers). We also verified similar levels of proteasome subunits among genotypes (Figure 1 for Reviewers). But, we do not exclude that persistent accumulation of p62 in their livers may affect the proteasome activity. Currently, we are investigating the phenotypes of the genetically modified mice with ageing. In the present study, we have focused on the property and function of p62-gels, and thus the phenotypic analyses are beyond of the scope of the present work.

205

Figure 1 to Reviewers

210 **Figure 1 to Reviewers**

(a) Immunoblot analysis. Homogenates from livers of three 5-week-old *HyD-LIR^{fllox/fllox}* and *HyD-LIR^{fllox/fllox}; Alb-Cre* mice were subjected to SDS-PAGE and immunoblot analysis with the indicated antibodies. Data shown are representative of three separate experiments. There is no significant difference of proteasome subunits including $\alpha 6$, $\beta 1$ and Rpt6.

215 (b) Peptide hydrolysis activity of 20S and 26S proteasomes. Homogenates from 5-week-old *HyD-LIR^{fllox/fllox}* and *HyD-LIR^{fllox/fllox}; Alb-Cre* livers were fractionated by glycerol density gradient centrifugation (10-40% glycerol from fraction 1 to fraction 30). Aliquots of each fraction were used for the assay of chymotryptic activity of proteasomes using succinyl-Leu-Leu-Val-Tyr-7-amido-4-methylcoumarin (Suc-LLVY-AMC) as a substrate without (top panel) and with
 220 (bottom panel) 0.05 % SDS. The sedimenting positions of 20S and 26S proteasomes are shown by arrows depicted '20S proteasome activity' and '26S proteasome activity', respectively. Note that whereas 26S proteasomes exist in active forms in tissues, 20S proteasomes are latent and activated artificially by a low concentration of SDS. Peptidase activity was measured using a fluorescent peptide substrate, Suc-LLVY-AMC, as described previously¹.

225 1. Komatsu M, *et al.* Loss of autophagy in the central nervous system causes neurodegeneration in mice. *Nature* **441**, 880-884 (2006).

Comment-5:

230 The other surprising finding of the current study is the fact that overexpression of HyD-LIR impaired selective autophagy, which affected several organelles in hepatocytes, yet no obvious liver defects were observed when compared to liver-specific *Atg7*^{-/-} deficient livers despite an increase in *Nrf2* activation. This let the reviewer to think that aberrant accumulation of selective autophagy cargo is harmless to the cells? Are the mitochondria normal? Do they respire normally? Is there any superoxide leakage?

235

Reply-5:

We measured the activity of succinate dehydrogenase in livers of 4-weeks-old *HyD-LIR-Venus*^{flox/flox}; *Alb-Cre* and age-matched control mice. The activity in the *HyD-LIR-Venus*^{flox/flox}; *Alb-Cre* mice was higher than that in control mice, probably due to the increased number of intact mitochondria (Figure 2 for reviewers).

240

At least, 4-weeks-old *HyD-LIR-Venus*^{flox/flox}; *Alb-Cre* mice did not show any pathological signs in the liver. We think that prolonged suppression of selective autophagy in their livers should cause severe liver disorders. Currently, we are investigating the phenotypes of the genetically modified mice with ageing. In the present study, we have focused on the property and function of p62-gels, and thus the phenotypic analyses are beyond the scope of the current work.

245

Figure 2 to Reviewers

Figure 2 to Reviewers

Mitochondrial succinate dehydrogenase activity per liver were measured. Data are means \pm s.e of 3 mice in each group, $**P < 0.01$ as determined by Welch's *t*-test. The SDH activity was assayed as described previously².

250

2. Ueno T, Watanabe S, Hirose M, Namihisa T, Kominami E. Phalloidin-induced accumulation of myosin in rat hepatocytes is caused by suppression of autolysosome formation. *Eur J Biochem* **190**, 63-69 (1990).

255

Comment-6:

Please include the controls (HyD-LIRflox/flox) for Figures 3g and 3h.

Reply-6:

260 EM images for the control (*HyD-LIR-Venus^{flox/flox}*) were included in the Figure 3i of the revised manuscript.

Comment-7:

265 *Figure 3f (left), need to include p62 and Figure 3 (right) need to include LC3. It is also suggested to give different letters for these two images.*

Reply-7:

According to the comment, we modified the Figure.

270 Minor comments:

*Page 9, line 282, change “increased” to decrease.
Abstract, line 43, change “distinct to” to “distinct from”*

Reply:

275 We regret the errors. The word has been corrected.

Referee #3

General comments:

280 *In this manuscript, authors shown P62 body is gel form by phase separation, moreover, they shown formation of autophagosome on p62 droplet. Finally, they shown recruitment of keep1 and activation of Nrf-2 on p62 droplet and the combination of Nrf2-activation with autophagy impairment cause liver damage.*

285 *The formation of p62 droplet by phase separation have been reported recently, however, the physiological significance of phase separation of p62 is not clear, by shown p62 droplet is the site of Nrf2 activation and selective autophagosome formation, authors provide important*

conceptual advance to the field of selective autophagy, it is a comprehensive study and for most part, data are compelling. I have a few suggestions to further improve this study.

290 Reply:

We really appreciate this reviewer for positive evaluation of our study. In accordance with the valuable comments, we performed several experiments to strengthen our conclusions.

Comment-1:

295 *Authors need to shown the recruitment of Keap1 into p62 droplet in vitro.*

Reply-1:

Thank you for your suggestion. We conducted the in vitro assay with recombinant proteins and found that Keap1 was recruited to p62-4xUb condensates and that the ternary condensates were
300 bound to Atg8-GUVs (Figure 4a and b in the revised manuscript).

Comment-2:

*Authors conclude Nrf2 activation on p62 gel by disruption of p62/keap1 interaction by chemical inhibitor and using p62 mutant which can't bind to Keap1, which are nice, however, this
305 reviewer argue these data is open to alternative explanation and does not provide a direct evident for Nrf2 activation on p62 gels. Authors should try p62 mutants which can't form droplet.*

Reply-2:

310 Thank you for your suggestion. Using an oligomerization-defective K7A D69A mutant of p62, which shows defective droplet formation, we found that liquid-droplet of p62 is indispensable for the activation of Nrf2 (Figure 4j-1 of the revised manuscript).

Comment-3:

315 *Does Atg1 or other autophagy machinery localized on p62 droplet? If so, it will greatly support author's conclusion that autophagosome form on p62 gel.*

Reply-3:

In addition to FIP200, WIPI2 and ATG16L, we carried out the immunofluorescence staining
320 with an antibody against ULK1 (yeast homologue of ATG1) and found the localization of ULK1 on the droplets (Figure 1d of the revised manuscript).

Referee #4

325 General comments:

In this manuscript, the authors show that p62-positive puncta within the cells are gel-like structures. The p62-gels are degraded by selective autophagy through their engulfment by autophagosome, which depends on p62 interaction with LC3/GABARAP proteins. Ultimately, the p62 gels serve as a platform for activation of Nrf2 through the sequestration of Keap1 into the p62 gels.

Major comments

Comment-1:

335 *The authors nicely show that in Huh-1, an hepatocellular carcinoma cell line expressing high level of p62, the p62 structures are gels. To rule out the possibility that the existence of p62 gels depends on p62 expression levels and/or it is cell type-specific, it is important show that p62 also organizes in gel in non-carcinogen (hepatic) cells expressing low or moderate levels of p62.*

340 Reply-1:

Thank you for this suggestion. This comment was also raised by the Referee 2 (Comment-1 of Referee 2). We have analyzed the p62-gels in Huh-1 cells, but also normal cells, mouse embryonic fibroblasts (MEFs) (Fig. 2 and Supplementary Figure S1a in the original manuscript). In addition to those cells, we investigated the p62-structures in mouse primary culture hepatocytes and another cancer cell line, HeLa cells. After treatment with arsenite, p62-structures were formed in these normal and cancer cells. Similar to the p62-gels observed in Huh-1 and MEFs, they were round and positive for phosphorylated forms of p62, Keap1, and ubiquitin. Thus, we concluded that the properties of p62-gels which we have observed in Huh-1 and MEFs are general. We presented the results (Supplementary Fig. S1b and c of the revised manuscript) and described the above-mentioned data (Page 5, lines 135-137 in the revised manuscript).

Comment-2:

355 *In the text (line 147) as well as in the Material and Methods, the authors refer to CLEM. None of the figures (Figure 2 and Figure S2), however, depicting EM data show the corresponding fluorescence images used for the correlation and the undoubtable identification of the structures of interest. The CLEM data have to be added to the paper since they will help to determine which structures have been solved at the ultrastructural level.*

360 Reply-2:

We have added the fluorescence microscopy – TEM correlation images as requested (Fig. 2 and Supplementary Fig. S2 of the revised manuscript). Please also see the response to the next comment below, emphasizing the unique TEM morphology of the p62-gels. We are able to identify the p62 gels even without correlation with fluorescence.

365

Comment-3:

-In line with the previous comment, in Figure 2 the authors claim that the EM micrographs show either p62 gel, phagophore containing p62 gels, or autophagosome that do not contain p62 gel. Without any labelling (immuno-EM and/or CLEM), based on which criteria the authors can establish which one of these 3 structures does contain or not p62?

370

Reply-3:

We can identify p62-gels in conventional EM images since they have a unique morphology, which is mentioned in Results (Page 5, lines 153-156 in the revised manuscript). The same unique morphology has been reported by another group earlier: Jakobi, 2020, Nature Commun. (Nat Commun 440, 2020, Figure 5c-d), which we also site in the text (Page 5, line 156, Ref. 30).

375

Comment-4:

How the authors can distinguish the morphology of a phagophore from an autophagosome especially using 3D reconstruction on serial sections that do not resolve entirely the complete structures? In particular, the authors cannot conclusively state that an autophagosome is an autophagosome if its entire structure has not been resolve as the opening, which will classify it as a phagophore, could be present in the part of the has not been resolved/imaged. The authors should consider to use focus-ion beam-scanning electron microscopy (FIB-SEM) in order to resolve the complete 3D-volume of their structures and therefore undoubtedly distinguish autophagosomes from phagophores.

385

Reply-4:

The reviewer is correct, if we do not see the whole volume in the 3D image, we cannot be 100% sure in all cases whether a structure is an open phagophore or a closed autophagosome (unless we see it is still open or closed). However, both a phagophore and an autophagosome indicate autophagic sequestration. Thus, we do not think this is a crucial issue for the conclusions of the paper. Nevertheless, we have carefully gone through the images and 3D models, and changed

390

395 the wording in the manuscript. In case we can see the structure is still open, we call it ‘isolation membrane/phagophore’. In case we do not see whether the structure is open or closed, we call it ‘isolation membrane/phagophore/autophagosome’.

Comment-5:

400 “On the basis of the morphological analysis, we concluded that there are two distinct ways of autophagosome formation on the p62-gels, either the p62-gel is engulfed by the forming autophagosome, or the autophagosome forms next to the p62-gel but does not engulf it”. EM are steady-state images, therefore the p62 gel not engulfed may just represent an earlier event, prior the subsequent engulfment. The 2 types of segregation probably correspond to selective
405 and bulk autophagy, this latter occur at basal level in numerous cell types. This has to indicated in the text.

Reply-5:

We fully agree and have added the requested sentence to Results (Page 6, lines 175-176 in the
410 revised manuscript).

Comment-6:

Lines 149-151 and lines 157-161, without quantification, these observations are highly
415 speculative.

Reply-6:

The statements were not speculative, instead, they were based on qualitative observations. However, we have quantified the structures and reworded the text in Results as follows:

1. 19.5% of p62-gels in *p62-GFP^{KI/+}* MEFs colocalized with WIPI2, an isolation membrane
420 /phagophore marker at 6 hr after removal of As[III] (Supplementary Fig. S1e) (Page 5, lines 142-144).
2. The results now reads: In agreement of immunofluorescence analysis with WIPI2 antibody (Supplementary Fig. S1e in the revised manuscript), the p62-GFP gels were frequently observed inside isolation membranes/phagophores and autophagosomes, which in many cases had multiple double membranes on top of each other around them (Fig. 2a-f in the
425 revised manuscript) (Page 5, lines 156-159 in the revised manuscript).
3. And further: Approximately 50% (49 out of 99) of the phagophores/isolation membranes and autophagosomes locating next to p62 gels were enveloping p62-gel, while the rest were enveloping other cytoplasmic components (Page 6, lines 165-168 in the revised
430 manuscript).

Comment-7:

Line 150 “They were frequently observed inside isolation membranes/phagophores and autophagosomes, which in many cases had multiple double membranes around them”. Only the
435 EM micrograph on Figure 2a show a clear “multiple double membrane”, while the rest of the micrographs does not. In addition, the enwrapping of an autophagic cargo by multiple double membranes have already been shown for mitophagy by one of the authors. This has to be mentioned in the text.

440 Reply-7:

The multiple double membrane was observed in some, but not all phagophores/autophagosomes. The Figures thus reflect the findings we made. We assume the Reviewer refers to Dev Cell. 2019 Sep 9;50(5):627-643. In this article we describe the formation of autophagosomes around damaged mitochondria. The formation of phagophores
445 initiated at several sites around the cargo mitochondrion simultaneously, and then these phagophore precursors fused together to form an autophagosome around the mitochondrion. These autophagosomes were thus lined by one double membrane (i.e., two lipid bilayers). This is different from what we observed in the present study. Part, but not all, of the p62 gels had several phagophores forming around them, on top of each other. If all of these phagophores
450 closed to form autophagosomes, the cargo would have several double membranes around it (i.e., four or more lipid bilayers). To make the difference between the present study and Dev Cell 2019 paper more clear, we modified the text as follows: ... autophagosomes, which in many cases had multiple double membranes on top of each other around them (Fig. 2a-f) (Page 5, line 158-159 in the revised manuscript).

455

Comment-8:

To confirm their conclusion that the engulfment of p62 gel by autophagosome is dependant of the interaction between p62 and LC3/GABARAP, the authors should show by IEM or CLEM that LC3 or GABARAP are present onto the forming phagophore surrounding the p62 gel.

460

Reply-8:

To address this comment, we have provided the following published or novel evidence:

1. LC3-positive puncta are co-localized with p62-structures (Cell, 131, 1149, 2007).
2. p62-structures consisting of LC3/GABARAP-binding defective p62 are not degraded by
465 autophagy (J Biol Chem, 283, 22847, 2008).
3. Phagophore profiles surround p62-body as shown by IEM (J Cell Sci, 128, 4453, 2015).

4. The p62-structures show liquid-droplet properties under stress conditions (EMBO Rep, 21, e48902, 2020).

In the present study, we showed that p62-structure are gel-like liquid droplets (Fig. 1) and that their autophagic degradation is impaired by the inhibition of the interaction between p62 and LC3/GABARAP (Fig. 3). On the basis of the published and novel evidence, we concluded that the engulfment of p62-gel by autophagosome is dependent on the interaction with LC3 and/or GABARAP (Page 8, lines 241-255 in the revised manuscript).

475 Minor comments

Comment-1:

It would be highly beneficial for the reader that the authors precisely define and then are consistent with the terminology such as body, liquid droplet and gel structure. In the current version of the manuscript, these terms are not always properly used and/or easily mixed.

480

Reply-1:

Thank you for this suggestion. We unified the terms to “gel”.

Comment-2:

485 *In the FRAP experiment (line 115 and Figure 1c), the authors observed that the fluorescence recovery time of the GFP-p62-positive structures is slower than the one generally obtained for liquid droplets, leading to the fact that then p62 structures are gels. Does the recovery time of the GFP-p62 gels correspond to the ones observed for others gels described in the literature?*

490 Reply-2:

The observation that fluorescence recovery time of the GFP-p62-positive structures is slow was already made by Sun et al. and Zaffagnini et al. (Refs 17 and 18). We cited both papers (Page 4, line 119 in the revised manuscript).

495 Comment-3:

Line 79, “... called LIR/GIM to interact with autophagosome-localizing ATG8 ...”, use a single nomenclature, LIR/GIR or AIM/GIM

Reply-3:

500 In the case of mammal, we prefer to use “LC3-interacting region (LIR)” and “GABARAP-interacting motif (GIM)” in accordance to the original researches (J Biol Chem, 282, 24131, EMBO Rep, 18, 1382, 2017)w.

Comment-4:

505 *Line 537, "... the cells were fixed in 0.1 M PB containing 4% paraformaldehyde and 4% sucrose... ", it seems that "sucrose" should be replaced by glutaraldehyde*

Reply-4:

This is the fixation used for immunoEM. 4% glutaraldehyde would destroy all epitopes, thus,
510 4% sucrose is correct.

REVIEWERS' COMMENTS

Reviewer #1 (Remarks to the Author):

The authors have addressed all my comments adequately and in my opinion the manuscript is suitable for publication in Nature Communications.

Reviewer #2 (Remarks to the Author):

This reviewer thank the authors for their responses to the critiques raised and for the additional data provided in the revised version.

Reviewer #3 (Remarks to the Author):

Authors had addressed my queries satisfactorily, congratulations for this beautiful work!

Reviewer #4 (Remarks to the Author):

The authors have respond satisfactorily to all my comments/requests. This high quality manuscript for me can be accepted for publication.

Reviewers' Comments: (italicized)

Comments

Reviewer #1 (Remarks to the Author):

The authors have addressed all my comments adequately and in my opinion the manuscript is suitable for publication in Nature Communications.

Reviewer #2 (Remarks to the Author):

This reviewer thank the authors for their responses to the critiques raised and for the additional data provided in the revised version.

Reviewer #3 (Remarks to the Author):

Authors had addressed my queries satisfactorily, congratulations for this beautiful work!

Reviewer #4 (Remarks to the Author):

The authors have respond satisfactorily to all my comments/requests. This high quality manuscript for me can be accepted for publication.

Rely

I thank all reviewers for the positive evaluation of our revised manuscript.